

# Automated classification of seismic signals recorded on the Åknes rockslope, Western Norway, using a Convolutional Neural Network

Nadège Langet[1] and Fred Marcus John Silverberg[2]

[1]NORSAR, Gunnar Randers vei 15, N-2007 Kjeller (Norway)
[2]The Centre for Earth Evolution and Dynamics (CEED), University of Oslo (Norway)

**Correspondence:** Nadège Langet (nadege@norsar.no)

**Abstract.** A Convolutional Neural Network (CNN) was implemented to automatically classify fifteen years of seismic signals recorded by an eight-geophone network installed around the backscarp of the Åknes rockslope in Norway. Eight event classes could be identified and are adapted from the typology proposed by Provost et al. (2018), of which five could be directly related to movements on the slope. Almost 60,000 events were classified automatically based on their spectrogram images. The

performance of the classifier is estimated to be close to 80%. The statistical analysis of the results shows a strong seasonality of the microseismic activity at Åknes with an annual increase in springtime when the snow melts and the temperature oscillates around the freezing point, mainly caused by events within classes of low-frequency slopequakes and tremors. The clear link between annual temperature variations and microseismic activity could be confirmed, supporting thawing and freezing processes as the origins. Other events such as high-frequency and successive slopequakes occur throughout the year and are

potentially related to the steady creep of the sliding plane. The huge variability in the annual event number cannot be solely explained by average temperatures or varying detectability of the network. Groundwater recharge processes and their response to precipitation episodes are known to be a major factor of sliding at Åknes, but the relationship with microseismic activity is less obvious and could not be demonstrated.

**Keywords.** Microseismic monitoring - Unstable rockslope - Classification - Convolutional Neural Network

## 1 Introduction

Norwegian landscapes have been shaped by deglaciation over thousands of years, creating very steep and mostly water-filled valleys, the fjords, over the western parts of the country. The steep flanks of the fjords are potentially unstable and their collapse poses a significant hazard. Most of these flanks are uninhabited and direct effects of a collapse would be rather limited. However, the mass sledging into the fjord can generate a local flood wave causing both infrastructure damages on

shorelines and casualties. During the 20[th] century, 175 victims have been reported in Norway due to such events (Kveldsvik, 2008). Worldwide, different factors contribute to the triggering of slides. For example, large earthquakes have the potential to trigger landslides or accelerate their sliding (Lacroix et al., 2015; Bontemps et al., 2020). Nevertheless, the most common causes for sliding are related to variations of the water content at depth both in terms of amount (e.g. infiltration following precipitation; Helmstetter and Garambois, 2010; Bontemps et al., 2020), pressure and state (between solid and fluid, i.e. during



freezing or thawing; e.g. Krautblatter et al., 2013; Blikra and Christiansen, 2014). In addition to its sharp relief, Norway experiences cold winters and mild summers, especially in the North, favourable for the temporary extension of frozen ground in winter or the conservation of permafrost. A permafrost probability map for the steep slopes of Norway was established by Magnin et al. (2019). 2% of Norwegian slopes are believed to be covered with continuous permafrost, 9% with discontinuous permafrost and up to 20% with sporadic permafrost. Elevations at which permafrost can be observed depend mainly on latitude

and slope aspect, i.e. a high latitude and an exposure toward north would lower the limit. It was additionally revealed that permafrost may occur on steep slopes at lower elevations than on less pronounced topography, particularly for north-facing flanks. Moreover, permafrost could have acted as a slope stabiliser over the past thousands of years. As a consequence of the current global warming, the permafrost cover shrinks, leading to rock slopes becoming more unstable (Hilger et al., 2021). Therefore, Norway may experience more catastrophic sliding events in the near future and a crucial aspect is to develop

automatic monitoring systems on the slopes to mitigate the related risk.

  Currently, only a few sites are instrumented in Norway (e.g. Blikra et al., 2005; Nordvik et al., 2010; Blikra and Christiansen, 2014; Böhme et al., 2013). The Åknes unstable rock slope, subject of the present work, started being heavily instrumented in 2004 (Nordvik et al., 2009) and is considered the best and most thoroughly monitored rock slope in Norway. As part of the monitoring system, seismic instruments were installed to passively record seismic activity on the slope. Because seismic

sensors do not only record events related to the movements of the slope, but also regional earthquakes and man-made noise, it is important to be able to distinguish between different types of signals. The network was fully functional in 2007 and since then, around 60,000 events were detected. The combination of this large amount of data and the need for real-time implementation then required automation of the classification task.

  In this paper, we will first present the Åknes site both in terms of geology and monitoring, with a particular focus on the

seismic monitoring, and then describe the automatic classifier implementation. The statistical analysis of the results shows the implications in terms of slope monitoring, particularly with the aim of better understanding the processes at play.

## 2   The Åknes unstable rockslope

### 2.1   Geology

  The Åknes unstable rock slope (62°10.77'N, 6°59.45'E) is located in the municipality of Stranda in the Møre og Romsdal

county in Western Norway (Fig. 1a). It overhangs the Sunnylvsfjord, which branches off to the Geiranger fjord in the South-East. This fjord became a World Heritage Site in 2005, attracting a growing number of tourists and cruise ships each year. The need for ensuring safety in the surroundings of this important touristic site partly explains why the Åknes slope received a great deal of attention. The slope faces south with a very steep topography from sea level to 1300 m altitude over a distance of about 1500 m (Ganerød et al., 2008). The rock formations consist of three main types of gneisses, which foliation is parallel

or sub-parallel to the slope. The steep slope angle (30-35°) and the foliation constitute the two main geological factors for instability. The unstable part is delimited in the West by a NNW-SSE striking and steeply dipping strike-slip fault resulting in a cliff of 10 to 40 m height (Fig. 1b). A NNE-SSW striking and gently NW dipping strike-slip fault forms the eastern boundary.

Earth **Surface**
**Dynamics**
Discussions

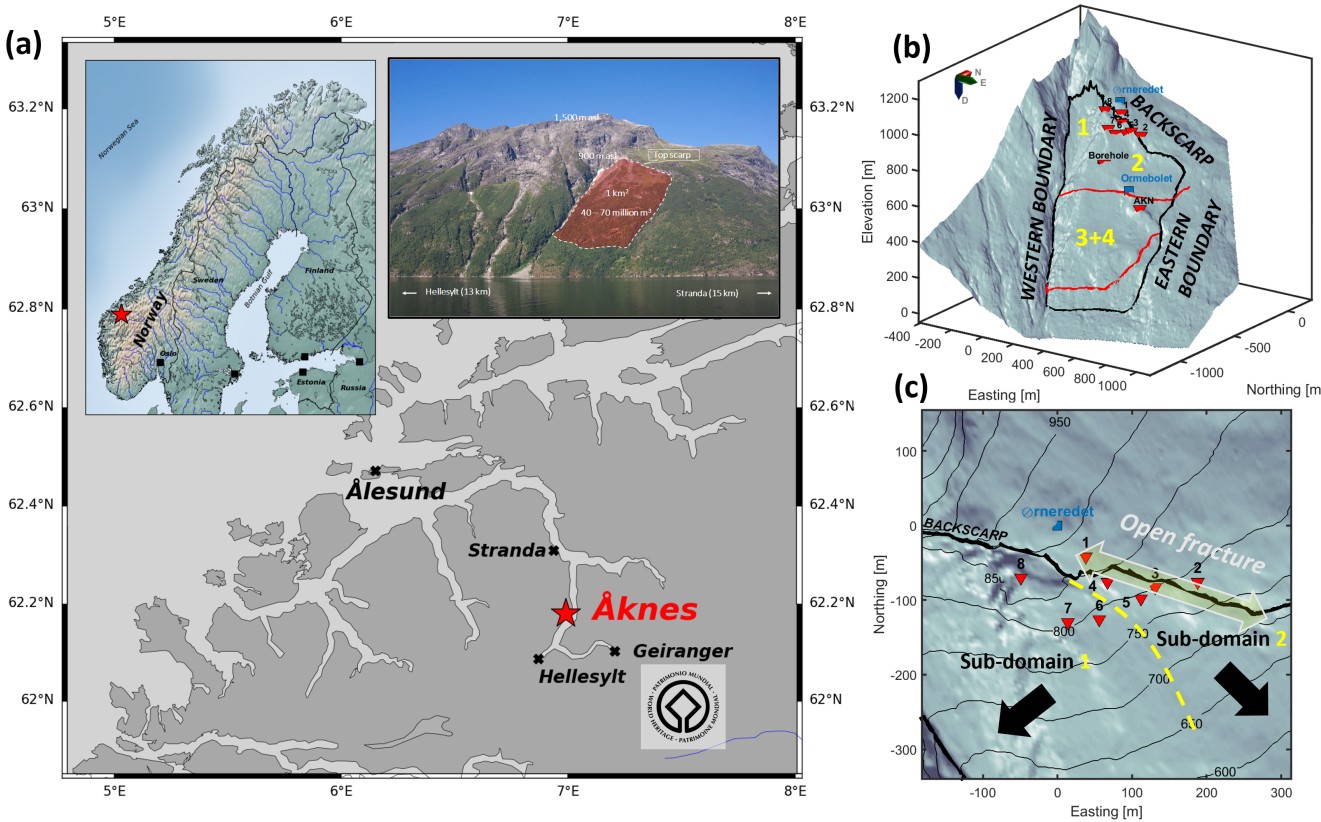

**Figure 1.** (a) Åknes rockslide location in Western Norway (red star). The photo on the top right shows the Åknes slope seen from the opposite side of the fjord (Roth and Blikra, 2010). The red shaded area represents the unstable area. Digital Elevation Model of the slope in 3D (b) and map view (c), zoomed to the top scarp, where the eight surface geophones are placed (red reversed triangles). The two main bunkers at the top and in the middle of the slope are displayed as blue squares. We use a local coordinates system centred on the upper bunker (Ørnereiret). In (b), the black contour delimits the whole area that could potentially collapse into the fjord. The red lines highlight two of the sliding planes. Sub-domains are indicated by yellow numbers. In (c), the backscarp splits the network in two, with geophones 1 and 2 placed on the stable ground behind the backscarp. The open fracture is highlighted by the yellow double-arrow. 50m-spaced elevation contours are also plotted. The two black arrows indicate the approximate displacement direction in sub-domains 1 and 2, and the yellow dashed curve marks the separation between the two sub-domains.





The upper limit, the back-scarp, is characterised on its western part by a cliff and graben, separating geophones 6, 7 and 8 from the others, and on its eastern side by an open fracture whose depth is difficult to estimate, but is estimated to be about
60 m deep in the western part, reaching to somewhat shallower depth towards the east (Fig. 1c). Geological observations at the surface and investigations using different geophysical methods (including radar, refraction seismics and resistivity profiles) led to the division of the unstable slope into four sub-domains (Ganerød et al., 2008), which - depending on the scenario - could collapse simultaneously or independently from each other. In the worst-case scenario, the total volume of unstable material was estimated to be more than 50 million m$^3$ (Fig. 1b), which would result in a tsunami with up to 85 m run-up height at the
nearby villages (Harbitz et al., 2014). Two sliding planes beneath the extensional upper sub-domains (1, 2) as well as beneath the lower compressional sub-domains were evidenced from surface observations (Fig. 1b, red lines).

## 2.2    Monitoring

First evidence of movement at the Åknes rockslope was reported as early as 1964, when locals observed a widening of the upper crack (Grøneng et al., 2011). Today, the slope is continuously monitored by NVE (the Norwegian Water Resources and
Energy Directorate) and is thoroughly equipped with numerous types of instruments. Displacement rates are measured by five extensometers within the upper crack, two lasers at the backscarp, ten permanent GPS points and thirty prisms covering the whole slope (Pless et al., 2021). Displacement measurements were also regularly performed through field campaigns of remote sensing techniques such as ground-based and satellite-based InSAR (Interferometric Synthetic-Aperture Radar; Kristensen et al., 2013; Bardi et al., 2016) and Terrestrial Laser Scanning (TLS;  Oppikofer et al., 2009). Annual displacement rates of
a few millimeters to up to 8 cm (Nordvik and Nyrnes, 2009) with an average of 1-2 cm/year were recorded (Ganerød et al., 2008). Larger displacements were observed in the upper part of the slope (sub-domains 1 and 2) compared to the lower part (sub-domains 3 and 4; Ganerød et al., 2008). Twelve boreholes were drilled and instrumented with inclinometers to monitor deformation as well as piezometers to survey groundwater level variations. Groundwater fluctuations are believed to be the main reason for the slope movement. Increases of the groundwater level are correlated with increases in measured displacement
rates (Nordvik and Nyrnes, 2009; Grøneng et al., 2011). The data are not self-sufficient, though, and need to be integrated with models of groundwater recharge processes. Hence, many hydrogeological field campaigns were conducted in summertime to map the streams on the slope and collect geochemical and flow data to better characterise water movement (Sena and Braathen, 2021). Last, but not least, a meteorological station installed close to the upper bunker (blue square in Fig. 1b,c) measures the temperature, precipitation, wind speed and snow cover.

As part of the monitoring system, seismic instruments were installed. The recorded data will be described in the next section. Passive seismic monitoring has the advantage of continuously recording waves propagating within the ground either in the form of well-identified seismic events or as background noise. In terms of slope monitoring, an increase in the number of detected seismic events can indicate movements of and on the slope. The magnitude of such events may help characterise
the type of slope movement, e.g. a few large events or many small events can result in the same amount of movement but caused by different processes. In addition, seismic events can be of various origins and they may be located at the surface





or at depth, hence providing information about the rock properties in the subsurface. Therefore, many rockslides worldwide are continuously monitored seismically (e.g. Spillmann et al., 2007; Dammeier et al., 2011; Lacroix and Helmstetter, 2011; Gomberg et al., 2011; Tonnellier et al., 2013; Arosio et al., 2018; Vouillamoz et al., 2018). Seismic precursors to rock slides

have been observed on different scales (Senfaute et al., 2009; Walter et al., 2012; Yamada et al., 2016; Poli, 2017; Butler, 2019; Zhang et al., 2019) and seismic records have been used to better understand rockfall dynamics as well (Hibert et al., 2017). In addition, an increasing number of studies leverages background seismic noise, which likewise contains valuable information (e.g. Colombero et al., 2018; Le Breton et al., 2021; Colombero et al., 2021). More precisely, tiny variations of seismic velocities can be measured by analysing long time series of ambient noise. It was observed that slope failures are

preceded by an irreversible drop in velocity, which can be directly linked to a loss of rigidity of the material. More interestingly, reversible and seasonal changes are generally measured, giving an insight on the different, site-specific processes at play.

## 2.3 Seismic data

At Åknes, a small-scale seismic network composed of eight 3-component geophones was set up in 2005 (Roth et al., 2006; Fischer et al., 2020) around the backscarp, which represents the most active zone, and registers permanently since 2006. The

network total extent is 200 by 300 m. Seismic data are recorded in triggered mode with a low threshold to include as many events as possible. In case of a trigger, 16 second-long files, sampled at 1000 Hz, are transferred to NORSAR in near real-time, where a more selective automatic STA/LTA detection (Withers et al., 1998) is performed. In 2009, the seismic instrumentation was supplemented by a broadband seismometer (AKN) in the central, more stable part of the slope for continuous data recording. This station, operated by NORSAR, provides data to the Norwegian National Seismic Network (NNSN) and the European

Integrated Data Archive (EIDA) and contributes to the regional and teleseismic earthquake monitoring. So far, it has not been used for slope monitoring in a systematic manner. Lastly, in 2017, a string of eight geophones was installed in a borehole in the upper part of the slope, just above one of the sliding planes (Fig. 1b). Although it would be interesting to integrate all microseismic records, the present paper will focus exclusively on the data registered on the surface geophone network. In section 5, we will discuss how the other available data streams may complement this work in the future.


It must be noted that network maintenance is challenging due to the harsh conditions and the limited field work season. In particular, cables between surface geophones and digitizer are the weakest links and are regularly sheared by either rockfalls or snow avalanches. For example, geophones 6, 7 and 8 in the western part of the slope are separated from the others by a gully, which is particularly prone to rockfalls, while geophone 2, in the eastern part of the slope, is placed in a snow avalanche

corridor. As a consequence, there are variations in detectability, which will be discussed later.

Rockslides can generate various types of seismic signals which have been analysed in several detailed studies at different sites (e.g. Helmstetter and Garambois, 2010; Occhiena et al., 2012; Tonnellier et al., 2013; Vouillamoz et al., 2018). Provost et al. (2018) brought together data recorded on different landslides (including Åknes) and proposed a standardised typology

reflecting the diversity of these signals, composed of three main classes (slopequakes, rockfalls and granular flows) subdivided



into several sub-classes. This typology is certainly not definite and must be adapted to site-specific characteristics, being aware that the network layout, the type of sensors and the acquisition parameters have an effect on records and data analysis and subsequently on the typology. Moreover, even though different types of signals are observed at various sites, their physical origins are still relatively unknown and can only be hypothesized. For example, the frequency content of a recorded signal de-

pends on the distance from the source to the recording station, especially if the medium is highly heterogeneous, dispersive or attenuative. Thus, records from a single event may have different seismic signatures at different stations. Likewise, two events having a similar source mechanism but occurring at different locations may display completely different waveforms. Event locations can help discriminating between source and path effects. However, locating events on rockslides is usually far from being straightforward. At Åknes, the medium is highly fractured and highly attenuative, hence, an adequate velocity model

is difficult to establish and seismic phases are not always easily identifiable on the waveforms. Several attempts have been made to locate the events, for example, Fischer et al. (2020) applied a back-propagation approach limited by the use of only a homogeneous velocity model and by constraining the locations to the surface. In the same paper, seismic events were classified into four categories (microseismic events, rockfalls, distant events and noise), but this work rather represented a feasibility study and was not systematically implemented as part of the automatic monitoring system.


The goal of the present work is (1) to identify the different signals recorded at Åknes, (2) to establish an automatic classifier, (3) to integrate it in the current processing routines in a way in which non-seismologists can exploit the catalogue of detected events without having to interpret the waveforms and (4) to discuss the classification of events in light of the processes involved in the slope movement. For that purpose, a large amount of waveforms was visually inspected and event classes were manually

assigned following a modification of the typology proposed by Provost et al. (2018). In the next paragraphs, various observed signals are described.

## 3   Description of observed seismic signals

The first main class of seismic events is called slopequakes. This terminology was introduced in order to avoid the wide variety of terms that can be found in the literature (microquakes, quakes, microseismic events, and so forth). This class is further

subdivided into simple and complex slopequakes. Simple slopequakes correspond to events that are believed to be associated to either the fracturing or the sliding of the slope. Based on their frequency content, several mechanisms may be hypothesized (Provost et al., 2018). Complex slopequakes encompass signals that are believed to be directly associated to phenomena affecting the slope stability at depth, such as e.g. tremors, but their origin is more speculative and so far, not so well understood.

At the Åknes site, high frequency (HF) and low frequency (LF) simple slopequakes are observed as well as tremors. HF slope-

quakes are impulsive and short-duration events, usually < 2 s (Fig. 2a). They contain energy at frequencies above 40 Hz and regularly comprise frequencies of 80 Hz or more. Differences in the higher frequency content may be an effect of the distance of the geophones to the source, therefore HF events are grouped into a single class. While lasting approximately as long as HF slopequakes, LF slopequakes feature emergent onsets hampering the identification of the signal start and their energy is mostly

Earth **Surface**
**Dynamics**
Discussions

**Figure 2.** Spectrograms and normalised waveforms of the maximum amplitude record of signals representative for each class: (a) HF slopequake, (b) LF slopequake, (c) tremor, (d) successive slopequakes, (e) rockfall, (f) regional earthquake, (g) spike and (h) noise. Note that colour scales are in a logarithmic scale [dB] and not normalised. Frequencies are only shown up to 150 Hz. In (f), both the P- and S-wave arrivals are visible at ca. 2 and 8 s, respectively.





concentrated in frequencies below 25 Hz (Fig. 2b). Usually, their maximum amplitude is much lower. Tremors have similar
characteristics to LF slopequakes but a longer duration (> 4-6 s, Fig. 2c). They partly exhibit a large portion of harmonic
signals. Lastly, we defined a class of successive HF slopequakes (Fig. 2d). They present roughly similar characteristics to HF
slopequakes but possess a longer duration (2 to 4 s) and consist of several bursts of energy (up to 4).

In addition to events taking place at depth, surface processes caused by the slope steepness and instability such as rock-
falls, snow avalanches and granular flows are recorded. Due to the file length limitation to 16 s and detection settings, snow
avalanches are very often either not detected or difficult to classify. Therefore, we do not have a dedicated class in the present
classifier, and the corresponding signals potentially are classified as noise (see section 5). Rockfalls, on the other hand, can
exhibit a wide range of waveforms and can be associated with different processes, e.g. a single rock splitting and falling down,
a rock bouncing repeatedly or a larger rock block collapsing. In general, rockfalls are characterised by emergent signals of
longer duration than simple slopequakes (> 2 s). They cover a large frequency bandwidth, reaching up to 150 Hz or more, and
their waveforms display several bursts of energy (Fig. 2e) as well as a high variability across the seismic network (Fig. A5).

Seismic instruments also record signals that occur farther away (e.g. regional earthquakes) and electronic noise, mostly as
spikes. Although not interesting to the slope monitoring, they need to be classified in order to be removed. Thus, spikes are
defined as short, narrow and high amplitude signals (Fig. 2g), whereas regional earthquake durations always exceed the file
length of 16 s (Fig. 2f). However, a pitfall for regional earthquakes is that depending on the source location, the S-wave may
not be visible on the records. Therefore, phenomena of longer duration such as snow avalanches or granular flows may be
considered by the classifier as regional earthquakes.

Lastly, we defined a "noise" class, corresponding to all signals that do not exhibit distinct features allowing them to be classified
as any of the previously described classes (Fig. 2h). In practice, this class only contains few events, since the STA/LTA detector
is sufficiently restrictive to discard noisy signals.

The main characteristics of the event classes are summarised in Table 1.

Hypothesized mechanical processes causing those events and their relationship to the slope geology are represented schemat-
ically in Fig. 3.

Vertical component waveforms of the example events in Fig. 2 recorded at all geophones are demonstrated in the appendix
(Figs. A1 to A8).

## 4 Classification

### 4.1 Background

Automated classification of seismic signals has started as early as in the 1990s, when Dowla et al. (1990) discriminated natural
earthquakes from nuclear explosions. Later on, it was successfully applied and developed in the field of volcano seismology
(e.g. Falsaperla et al., 1996; Ohrnberger, 2001; Masotti et al., 2006; Curilem et al., 2009; Langet, 2014; Maggi et al., 2017;
Malfante et al., 2018), and recently on rockslides (e.g. Hammer et al., 2013; Provost et al., 2017; Feng et al., 2020; Lin et al.,
2020). Amongst the most commonly used methods are Neural Networks (NN), Hidden Markov Models (HMM), Support



**Table 1.** Summary of the seismic event classes identified at Åknes. Asterisks (*) indicate events that occur directly on the slope and are therefore relevant and meaningful for the monitoring. The last column gives hypotheses on the physical processes causing the events, inferred both from the literature and the observations at Åknes developed in section 5. Probability density functions of the main signal characteristics are shown in Fig. A18 in the appendix.

| Class | Subclass | Duration | Frequency | Other | Potential origin |
|---|---|---|---|---|---|
| **Slopequake*** (at depth) | High-frequency (HF) | < 2 s | > 40 Hz, up to 80 Hz | very impulsive | shearing on sliding plane |
| | Low-frequency (LF) | < 4 s | < 25 Hz | lower amplitudes | fracture opening; new cracks (?) |
| | Tremor | 4-10 s | < 25 Hz | may contain harmonics | water flow in cracks (thawing) |
| | Succession of HFs | 2-4 s | > 40 Hz | several bursts of energy | shearing on sliding plane |
| **Rockfall*** | | > 6 s | large bandwidth, up to 150 Hz | several bursts of energy | rainfall, snow melt, ... |
| **Other** | Snow avalanche* | >16 s (file length) | dependent on avalanche regime (out of the scope of this work) | | |
| | Regional earthquake | | <10 Hz | | tectonics |
| | Distant, large rocklide / avalanche | | dependent on event size and distance to seismic sensors | | |
| | Noise | | dependent on type of noise; irrelevant for slope monitoring | | anthropogenic, electronic |

Vector Machines (SVM) and Random Forest (RF).

A classification problem is a supervised learning problem. This implies that a set of well-known data was labelled and used to train the machine learning algorithm. Building a good training set is crucial, since in the case of supervised learning, the system
is not able to learn anything new by itself: a signal will always be classified into one of the pre-defined classes. A training set should therefore be representative of the data and provide an exhaustive sampling in terms of (1) classes present in the data and (2) data quality. The representativeness is important due to the trade-off between the classifier generalisation and its precision, the aim being to maximise the number of correctly classified events, while minimizing the number of incorrect classifications for all classes.

For the Åknes dataset, events were classified by visual inspection of a large number of waveforms recorded in different years. Except the sparsely populated noise class (only 10 events), each class contains approximately 200 events. The distribution of classes across the training set is recognisable in Fig. A11a in the appendix. Most classes constitute around 12% of the training set. The class of HF slopequakes is twice as large in size (ca. 24%) as it appears to be the dominating event class in the whole dataset. The regional earthquake class contains slightly more events (16%) and was cross-checked with a reviewed seismic
bulletin (NORSAR, 1971) to ensure the proper labelling of corresponding events.



**Figure 3.** Sketch of the Åknes rockslope featuring local and distant natural phenomena generating seismic signals (see Table 1). Adapted from Fig. 10 in Ganerød et al. (2008) and Fig. 1 in Provost et al. (2018). Waveforms at the top are the same as the ones in Fig. 2.

## 4.2 Convolutional Neural Network

Machine learning methods developed quickly during the past few years. Breakthroughs occur not only due to the development of new conceptual approaches, but also because of huge improvements in hardware and computational capacity, which allow dealing with a much larger amount of data. Especially, deep learning methods are now able to classify images (Hinton et al., 2006) by creating a model not only for conditional probabilities $P(Y|X)$ (i.e. probability of an event to belong to class $Y$ knowing its set of features $X$), but for joint probabilities $P(X,Y)$ (i.e. the event belongs to the class $Y$ maximising the condi-





tional probability $P(Y|X)$ given its image $X$).

Traditionally, in machine learning methods such as e.g. Support Vector Machine (SVM) or Random Forest (RF), the input
data are discretised into feature vectors for which the features are extracted from the raw data with the aim to offer a best possible description. Due to the complexity of seismic waveforms, these features should describe the signals in the time domain in terms of duration, shape (e.g. number of peaks, peakedness,...), in the frequency domain (e.g. predominant and central frequencies, spectrum bandwidth, ...) and in the cepstral domain (the cepstral domain results from the inverse Fourier transform of the logarithm of a Fourier spectrum, e.g. Malfante et al., 2018). The number of features effectively used by the system may be re-
duced by removing correlated features and performing Principal Component Analysis (PCA). Feature-based methods have the advantage that they are easier to understand in the sense that features that are the most discriminant for each event class can be identified. However, their main drawback is that these features must be computed automatically, i.e. they rely on user-defined thresholds or computation parameters that should be suitable for the largest amount of data possible. Errors and uncertainties inherent to the feature extraction are fed into the automatic classifier and propagated. An example is the estimation of a signal
duration, which is usually not straightforward, since e.g. the level of noise can vary temporally and lead to an over-estimation of the signal length. Spectral features are usually extracted in the estimated signal duration window and may thus be affected. Altogether, the accumulation of errors and uncertainties may result in misclassification.

Thus, waveform complexity may act as an obstacle to classification, i.e. the classifier will be difficult to generalize and will
have a tendency to over-fitting, resulting in each individual event defining a class if the waveforms are not sufficiently similar. Spectrograms, on the other hand, constitute a simplified representation of waveforms in time-frequency domain. They contain the same information, but include a temporal reference in addition. Instead of resulting in a vector of discretised features, they represent a matrix of features that are not defined explicitly. For example, the duration information corresponds to the time span in which most of the energy is concentrated, instead of being defined by a scalar value expressed in seconds. Such information
matrices can be processed as images by Convolutional Neural Networks (CNN). Filters in the CNN's convolutional layers act as features; the more layers, the more filters are applied and the more features are extracted. The filter parameters are learnt during the training process. Pooling and resampling layers do not learn new parameters but decrease or increase the number of features. They may help in the identification of dominant features. The classification itself is carried out by a fully-connected layer. Convolution being a linear operation, convolutional layers must be accompanied by an activation function for the system
to learn non-linear features. There are many different activation functions available; for example, ReLU (Rectified Linear Unit) is widely used and consists in keeping only positive values after the convolution operation, while negative values are replaced by zeros. The architecture of the CNN, i.e. the number of layers, their order, the type of activation functions, etc., offers great flexibility and choices may affect the classification results. Still, well-known architectures have proved to be successful in a wide range of applications (e.g. LeNet-5, AlexNet, ResNet).
In the Åknes network, each event is recorded by eight geophones on three channels, i.e. on 24 traces in total. Depending on the source location and its distance to each geophone, wave onsets are not simultaneous across the network and the different ray





paths through a very heterogeneous medium distort the waveforms recorded at each geophone. However, since inter-station distances are short, differences in travel-times are expected to be small. Therefore, we simplify the information by (1) computing individual spectrograms for all channels and (2) stacking them altogether, resulting in a single image. Note that we only

stack spectrograms for geophones that detected the event and remove low-quality traces based on a user-defined threshold. This implies that the number of employed traces in each stack differs between events.

Practically, processing was implemented using the PyTorch python package (Paszke et al., 2019). We tested several CNN architectures (Silverberg, 2020) and chose the AlexNet architecture (Krizhevsky et al., 2017). In this pre-trained network, the

RGB spectrogram images must have a fixed size $[l,m,n]$ where $l$ is the number of channels (3 for RGB) and $m$ and $n$ are the height and width of the image, respectively. We computed spectrograms for the whole file length (16 s) using a 1 s-long sliding window with a 95% overlap. We truncated the spectrograms to 150 Hz, since only few events contain frequencies exceeding this value. After performance tests, the "afmhot" matplotlib sequential colormap was used to produce the spectrogram plot before its conversion to an RGB image of size $[3,224,224]$. The RGB values ranging from 0 to 255 are normalised for each

channel. The best-performing optimiser as well as other determinant parameters for the CNN, such as the learning rate, were determined in advance by both grid-search and cross-validation approaches. Finally, once the set of parameters yielding the best performance was found, the training of the final CNN was carried out on the entire training set.

### 4.3 Near real-time implementation

The supervised automatic classifier was implemented in near real-time. For each newly detected event, spectrograms are extracted, stacked and fed into the classifier. The resulting class is included in the catalogue of detections, a figure is produced (Fig. 4) and both are published on a dedicated webpage (NORSAR, 2022) with 10 to 15 minutes delay. Such figures are useful to visualise the event. In addition, the classifier returns probabilities, indicating the confidence with which the event was classified and, in case that the event fits multiple classes, the proportions with which these classes overlap. In Fig. 4, we show the

example of a HF slopequake classified with a probability of 100%.

In addition, the automatic classifier was applied to the database of past events. Note that we excluded year 2006, since the recording system was still in its test phase, and different sampling rates and file durations were used. Hence, the catalogue starts in 2007. Until the end of year 2021, a total of 59,608 events was classified. In the next section, we present the results

from the classification process and attempt to interpret the outcome for the purpose of slope monitoring.

### 4.4 Results

Figure A12 in the appendix shows the distribution of the automatically defined classes for the whole dataset. 20% of the events are not related to movements of the slope and represent regional earthquakes, electronic spikes and noise. Note that in compliance with the training set, the noise class is almost non-existent. 75% of the remaining 80% of events corresponds to



Earth **Surface**
**Dynamics**
Discussions

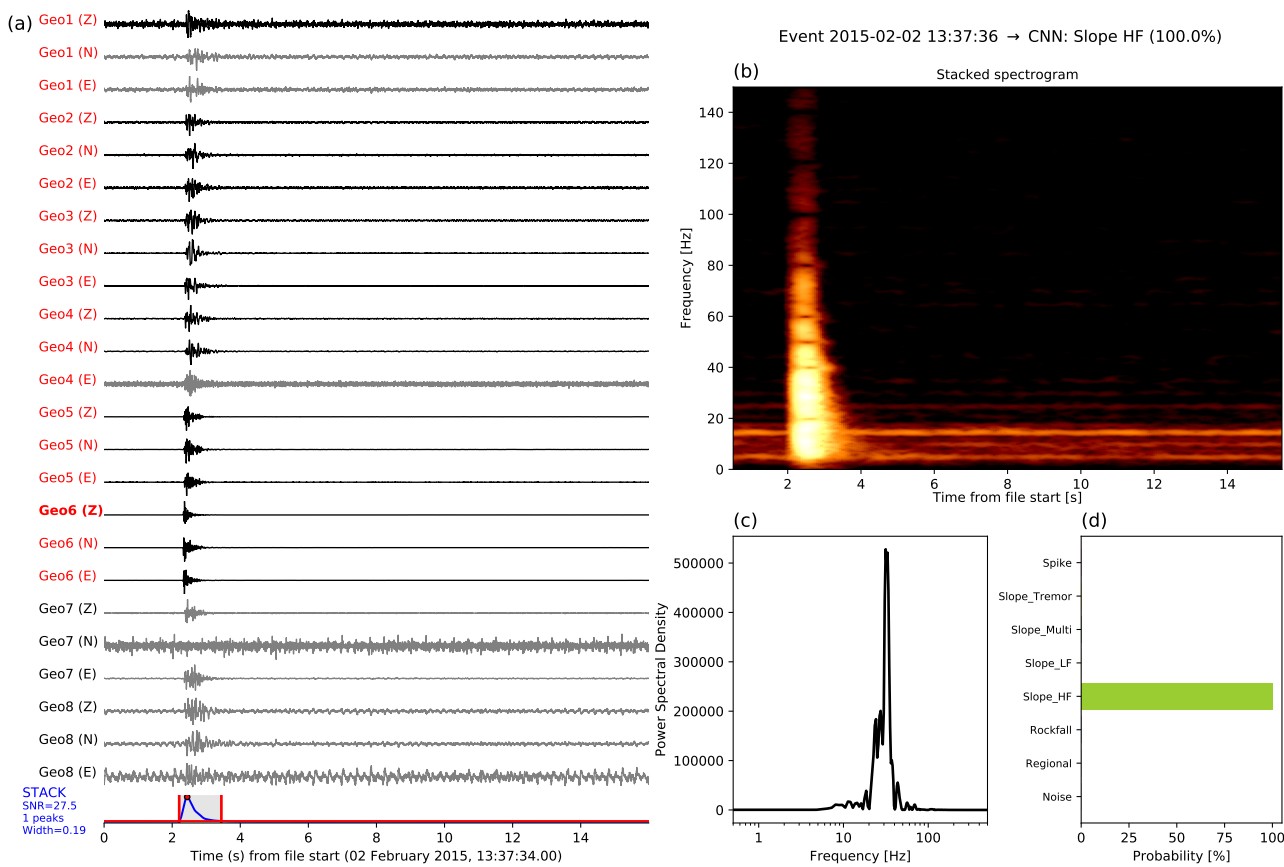

**Figure 4.** Example of an event classified as a high frequency (HF) slopequake by the automatic classifier. In (a), waveforms are shown for each component. Grey colour indicates geophones not contributing to the stack. Bold red text highlights the trace with maximum amplitude. The bottom trace represents the envelope stack of the traces with detection (black traces only). In (b), the spectrogram resulting from the stack of individual spectrograms computed for each channel is plotted employing the colour scale used in the CNN. In (c), the Power Spectral Density (PSD) of the trace featuring the maximum amplitude is illustrated. Finally, in (d), the bar displays the classification probability.



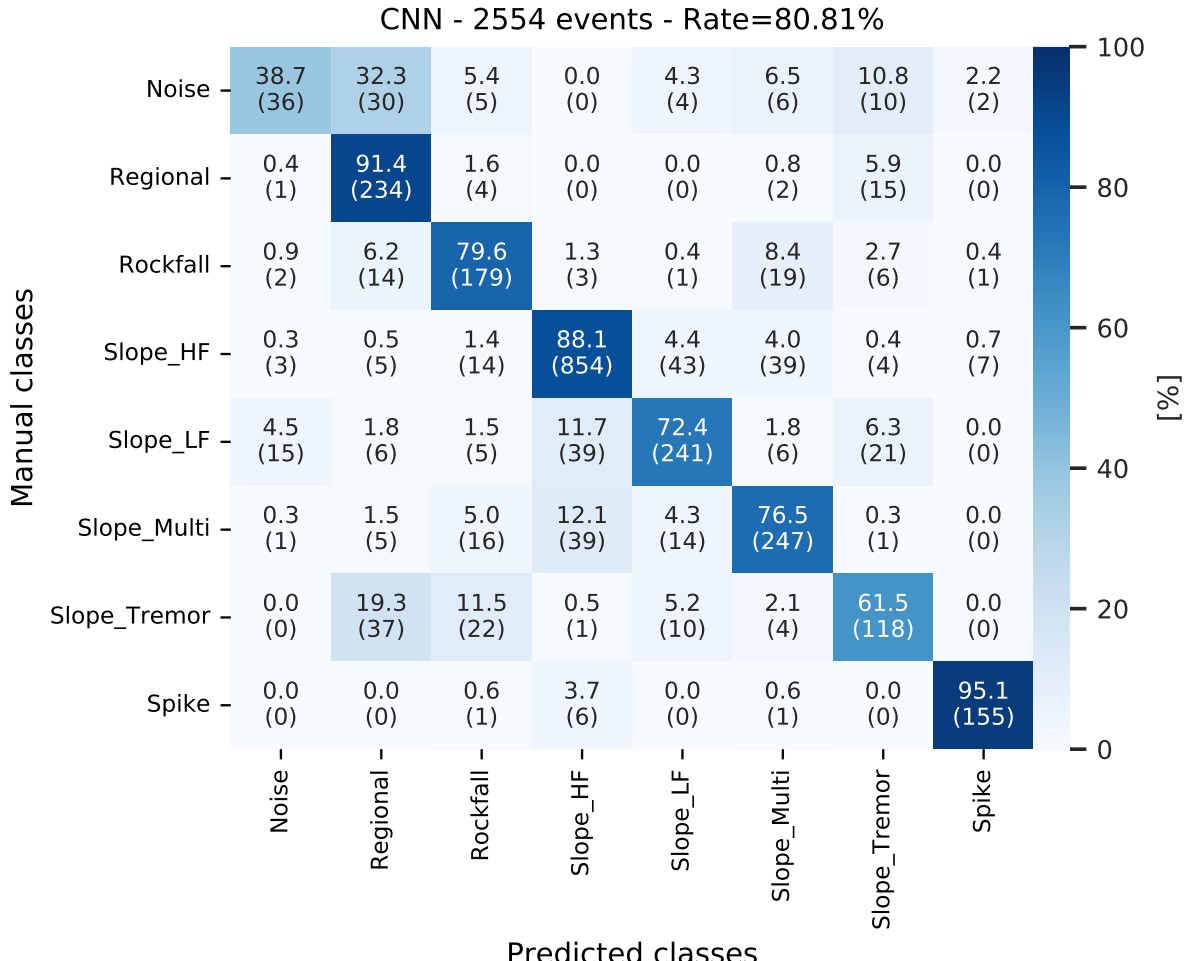

**Figure 5.** Confusion matrix comparing the results of manual and automatic classification for the test data set. The matrix is colour-coded by percentage rates. Numbers in brackets indicate the absolute number of events in each class.

slopequakes. Of those, HF slopequakes are largely predominant.

Since in addition, events are classified manually on a daily basis by visually scanning waveforms and spectrograms, we established a test set consisting of 2,554 events. Figure 5 illustrates the performance of the CNN classifier in the form of a confusion matrix for the test set, compared with the manual classification.

The distribution of manually-attributed classes within the test set is shown in Fig. A11b and we assume it to be representa-
tive of the whole dataset (Fig. A12). It is worthwhile noting that despite clearly defined characteristics, manual labelling is not straightforward for most of the events. Thus, manually defined classes should not be used as absolute ground truth and results



should be interpreted with caution.

The CNN's overall success rate is good (slightly above 80%) in spite of large discrepancies in success between classes.
Spikes are generally well identified as well as 79% of rockfalls. HF slopequakes are identified with a rate close to 88%,
whereas LF slopequakes reach 72%. More than 75% of successive slopequakes are properly classified. Tremors constitute the
event class that apparently is most difficult to classify with a success rate of only 61.5%. This is not surprising, since the char-
acteristics of this class are less clearly defined. Mostly, tremors are confused with regional earthquakes (19%) and rockfalls
(11.5%). Finally, less than half of the signals corresponding to noise are properly recognised by the CNN. This is expected,
since noise examples are severely under-represented in the training set; instead the system tends to classify noise as regional
earthquakes (32%). The classifier has a tendency to classify more regional earthquakes and HF slopequakes than suggested by
the manual classification, hence higher classification rates are reached for these classes.

To summarise, Fig. 5 shows that the classifier is able to properly discriminate events that do not occur on the slope with
the possible exception of noise. However, the number of noise records is only marginal and will not affect the interpretation
of the results. Events occurring on the slope reach classification rates of 75% on average. However, due to the complexity of
manual classification and the overlap between classes, those rates are considered satisfactory. The automatic processing results
are more homogeneous and consistent than the manual ones. For further interpretation, the automatically generated catalogue
was revised manually to delete records of electronic spikes erroneously classified as events.

## 5 Discussion

The initial motivation for building a classifier was to better characterise seismic events recorded at the Åknes site. Although
the number of detections provides information on the seismic activity rates over time, it does not allow to discriminate between
different mechanisms at play and may potentially be biased by including signals that are not linked to the slope. This is illus-
trated in Fig. 6 where the number of considered events drops to 47,561 after regional events, spikes and noise are removed,
thus comprising approximately 80% of detections. The trend of the cumulative number of events curve, however, does not
change much once these signals are removed. In general, we observe a deceleration of the seismic activity since mid-2012,
and regular inflections in the first half of each year, corresponding to an increase in seismic activity. In addition, Fig. 6 shows
the cumulative energy of these local events in terms of squared amplitude used as a proxy for magnitude. Together with the
cumulative number of events, it indicates whether the release of seismic energy was due to a few larger or multiple smaller
events occurring within a short period of time. No particular episodes of larger amplitude events emerge, suggesting that the
energy is generated by small, but numerous and regular seismic events.

In the previous work by Fischer et al. (2020) analysing the same dataset for the time period 2006-2013, a seasonality of
seismic activity was observed and a clear correlation with temperature variations was demonstrated. Both this study and the





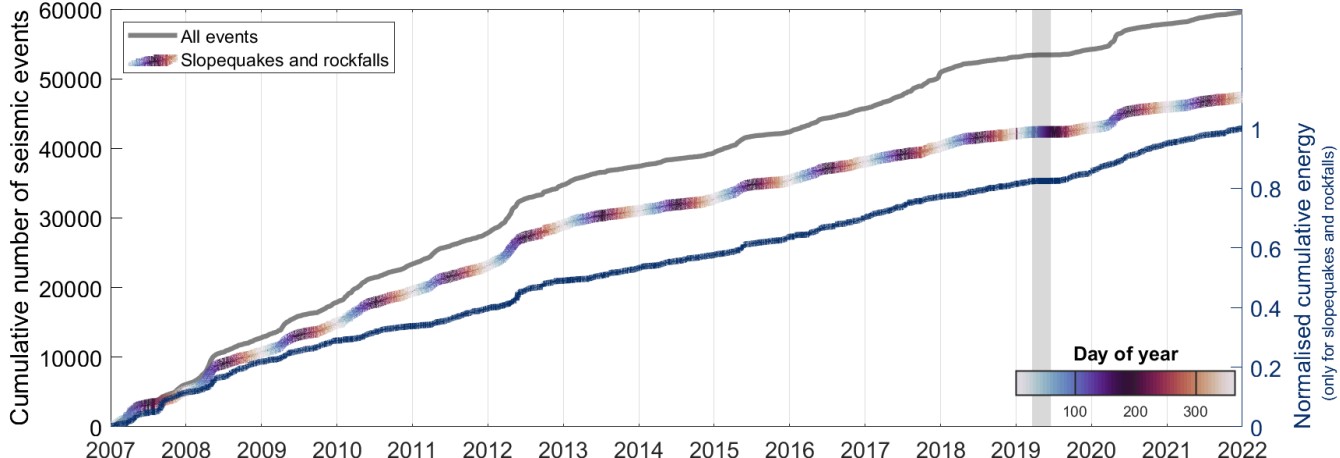

**Figure 6.** Cumulative number of seismic events and energy over time. Left axis: cumulative number of events including all detections (grey line) and restricted to events occurring on the slope (coloured line). The colour-code highlights the seasonality. Right axis: normalised cumulative energy (dark blue line). The time range in 2019 shaded in grey corresponds to a period when the system was down and no data could be collected.

work by Grøneng et al. (2011) using displacement data showed that there is an acceleration of the sliding during spring which could be attributed to snowmelt. Indeed, by melting, the snow facilitates the infiltration of water into the fractures and towards the sliding plane, in turn increasing the pore pressure and reducing the friction, and ultimately allowing for more sliding. In the following, we compare the catalogue of microseismicity with available meteorological data and expand the analysis further by adding the information on event types provided by the automatic classifier in order to identify which types of events are

more related to meteorological conditions. Figure 7 represents the monthly distribution of microseismic events for the different classes. Throughout a typical year, a significant increase in activity is observed in spring (from March to May), while summer months (June to September) are quieter. It is interesting to note that there is actually a gradual increase in seismic activity from October to February. This variation can be attributed to climatic conditions with temperatures below 4°C being reached regularly as early as in October. Temperatures are usually sub-zero from mid-November until the end of March. The transition

to temperatures above zero occurs in April, and generally, temperatures exceed 4°C by the end of May. Liquid water has a density varying with temperature. Most notably, under normal atmospheric pressure conditions, its highest density is reached at a temperature of 4°C, while water turns into solid ice at temperatures of 0°C or below. Ice is thus less dense than liquid water, which means its volume grows. If the cracks in the rocks are filled with water and the temperature is oscillating between 0°C and 4°C as observed from March to May, the variations in water density lead to seismic signals, which can be attributed

to either (1) water state changes corresponding to freezing and thawing in water-filled micro-cracks, (2) the creation of new cracks or (3) both. These explanations are in good agreement with the hypothesized mechanisms for the class of LF slope-quakes and, interestingly, their number increases in spring (Fig. 7) while decreasing from June to October. Relative variations

Earth **Surface**
**Dynamics**
Discussions

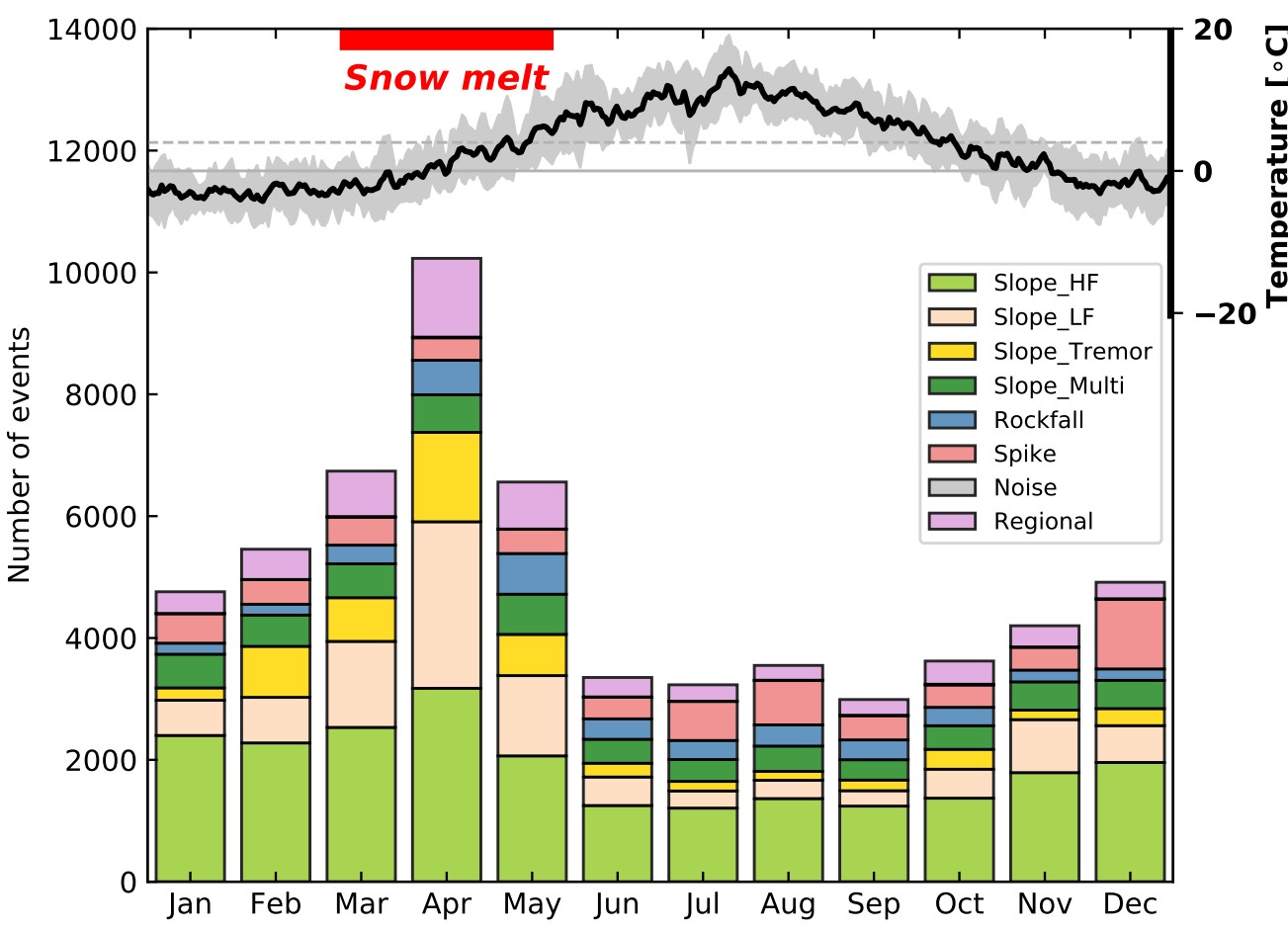

**Figure 7.** Seasonality of events and temperature per month for 15 years of data (2007-2021). Left axis: number of events. Bars are subdivided into event classes; noise records are not visible, since sparse. Top right axis: average temperature (black line) including standard deviation (grey shade). Solid and dashed horizontal grey lines indicate temperatures of 0 and +4°C, respectively.



**Figure 8.** Distribution of detected events over hours of the day for 15 years of data (2007-2021). Left axis: number of events. Bars are subdivided into event classes, see legend of Fig. 7 for explanation. Top right axis: relative variations of temperature ($\Delta\text{T}/\Delta\text{T}_{max}$) throughout the day, subdivided into seasons. For details, see Fig. B1.

of the number of events are displayed in Fig. A13 in appendix and show that in addition to LF slopequakes, tremors, rockfalls and, more surprisingly, regional earthquakes are also more numerous in spring. Rockfalls are more frequent in late summer

and beginning of autumn as well.

As evidenced in Fig. 7, temperature seems to be one of the predominant factors influencing the microseismic activity on the slope. This is also visible in Fig. 8 representing the distribution of classified events over the hours of the day. We observe slightly more activity during daytime (from ca. 8 am to 4 pm). This could indicate a potential contamination by anthropogenic

noise; however, the Åknes site is remote, only accessible by helicopter, and therefore protected from such sources of noise.





All human activities on the slope related to research and monitoring cease from end of October to at least end of April due to weather conditions prohibiting field work. Moreover, Fig. 8 reveals that all classes of events, except tremors, occur in relatively constant ratios independently of the hour of the day. Since tremors are rarely observed in summertime (Fig. 7), an anthro-pogenic origin is improbable. Our observations show that tremors are mostly recorded during the snowmelt season and during

daytime. The curves of relative temperature variations clearly indicate that daily variations are larger in spring and summer compared to autumn and winter (Fig. 8). This is expected, since at the latitude of Åknes, the daylight period is as short as five hours in winter. In Fig. B1 in the appendix, we show both absolute temperatures and their relative variations for each month of the year. On the whole, those observations strengthen and confirm the thawing-freezing cycles as the leading cause for tremors. Although LF slopequakes are more numerous in spring (Fig. 7), there is no obvious relation to daily temperature

changes (Fig. 8). Therefore, they could rather be related to either water infiltration and flow within pre-existing cracks, to the fracture opening or to the formation of new cracks.

In the absence of magnitude computation, event size is estimated by calculating the mean of the maximum amplitudes at each geophone. Amplitudes of slopequakes and rockfalls are plotted against time in Fig. 9. We observe a seasonal variation

for all classes (Fig. A14), notably that lower amplitude events ($\leq$150) are mostly recorded in autumn and winter, while higher amplitude events ($\geq$1000) are recorded throughout the year (Fig. A15). This is explained by a better detection capacity of the network from late October to May potentially due to an absorbing snow cover, enhanced geophone coupling and, although al-ready low, reduced anthropogenic noise. Further, the amplitudes of LF slopequakes and tremors are significantly lower than HF and successive slopequakes, as illustrated in Fig. A16. Rockfalls feature the highest amplitudes among slope-related events.

Interestingly, rockfalls are more frequent during certain periods of the years (Fig. 9e, Fig. A13). No obvious correlation of event amplitudes to precipitation could be established. More generally, although precipitation is commonly considered as one of the main causes of landslides (e.g., Helmstetter and Garambois, 2010) and at Åknes (Pless et al., 2021; Sena and Braa-then, 2021), a link between observed microseismicity and precipitation has not yet been fully evidenced for the Åknes site. This was also one of the results from the previous study by Fischer et al. (2020). The analysis is obfuscated by dependen-

cies on the speed of water infiltration and the process of groundwater recharge after a rain episode, which is varying across the slope (Sena and Braathen, 2021). More detailed analyses are required, e.g. by following an approach similar to Helmstet-ter and Garambois (2010) in which a cumulative rainfall index was defined to take into account the drainage of water over time.

As visible in Fig. 9a, the seasonal temperature variations are relatively stable over the years, in average not exceeding 1.5°C.

Nevertheless, a decrease in seismic activity is observed since 2012 (Fig. 6), illustrated more clearly in Fig. 10. The number of events halved between 2007-2012 and 2013-2021. Note that in 2019, the network was not functional from March to mid-June. The detection capacity depends of course highly on the number of functional geophones, but not only. Nevertheless, 2012 was the most active year so far, despite two geophones being out of order. In addition, much fewer events were recorded in 2013 and 2014, although the same geophones were functional. The difference in event numbers cannot be solely explained by

temperature variations, nor differences in precipitation. This suggests that while temperature and seismic activity are clearly

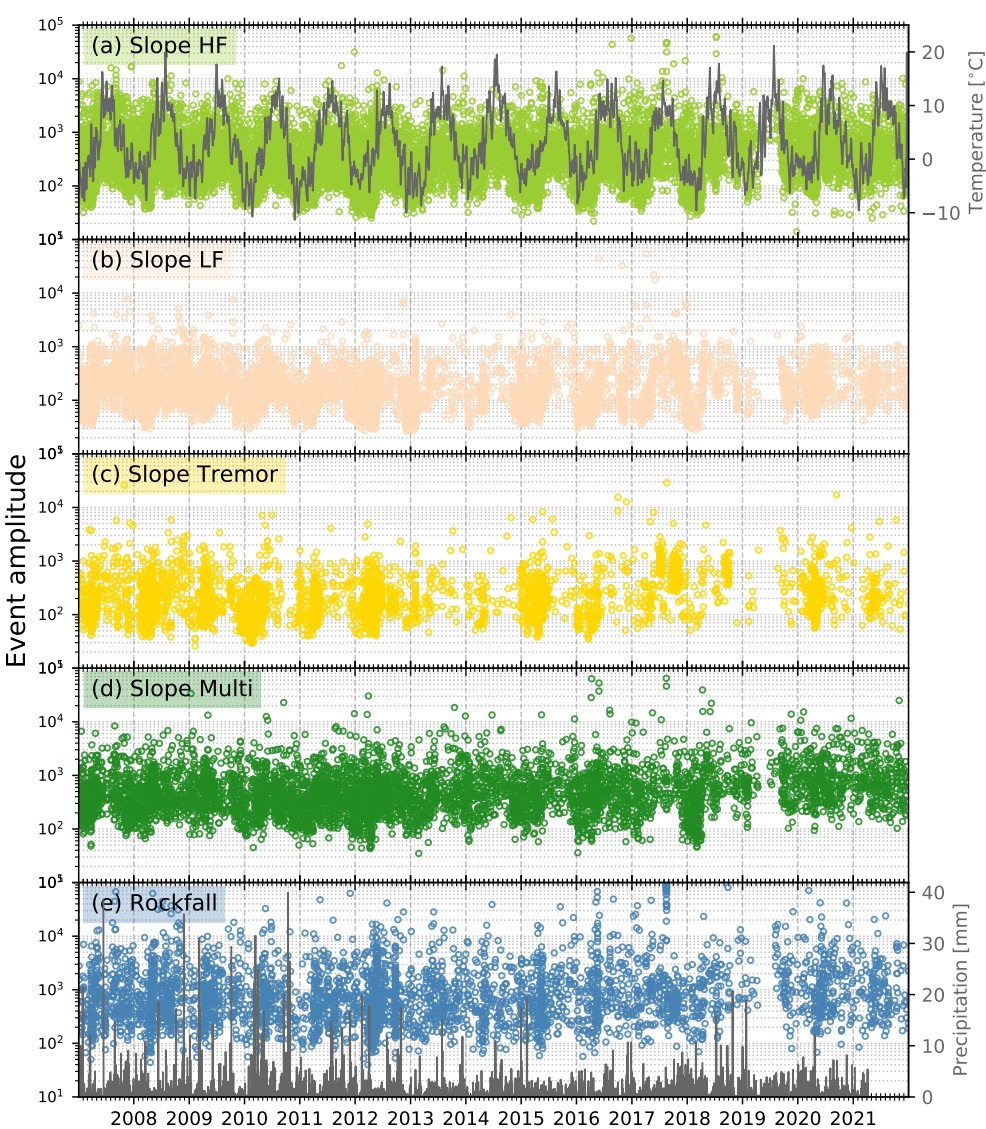

**Figure 9.** Amplitudes of events occurring on the slope over time. Temperature smoothed with a 7-day sliding window is added to (a). Precipitation is supplemented in (e).

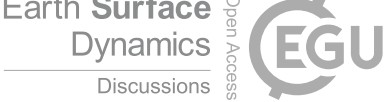

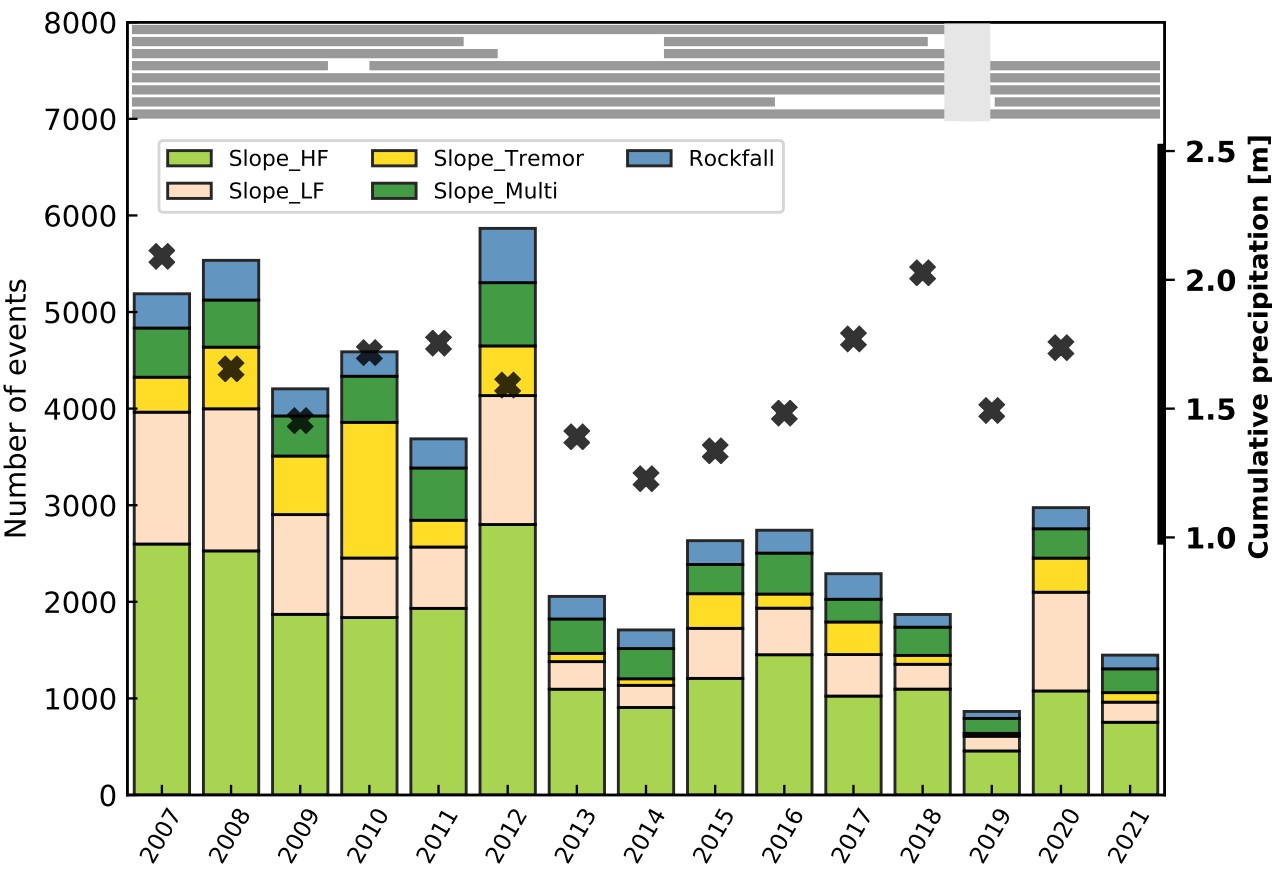

**Figure 10.** Distribution of slope-related microseismic events over years. Left axis: number of events. Right axis: cumulative precipitation for each year (black crosses). Note that the rain gauge malfunctioned in 2021, hence the value was removed. Top: data availability for each geophone (1: bottom to 8: top). Grey colour: available data; white: no data for specific geophone; light grey: network not functional.





linked, their relationship is more complex and additional factors need to be considered.

The file length of 16 seconds highly limits the capability of the classifier and could potentially explain the apparent seasonal increase in regional events (Fig. 7, Fig. A13). Since such a behaviour is not expected for earthquakes, it indicates that snow
avalanches, granular flow and landslides in the region surrounding the Åknes site are mistaken for regional events. This is exemplified in Figs. 11, A9 and A10.

In Fig. 11, we show an example of a regional earthquake and an event on the eastern part of the slope recorded on the broadband seismometer (AKN). In the absence of witnesses, it is difficult to distinguish if the event represents a snow avalanche or a granular flow. The corresponding records of the eight geophones are shown in Fig. A9 and A10 in the appendix. Although
in Fig. 11, the signal characteristics are unequivocal, e.g. with P- and S-wave arrivals clearly identifiable for the earthquake, the 16 s-long signals are too short to distinguish between both origins. For the earthquake (Fig. A9), only the P-wave onset was recorded, while the beginning of the trace in Fig. A10 can be misinterpreted as weak P- and stronger S-wave arrival at around 2 and 7 s, respectively. To avoid such misinterpretation, the broadband records could be integrated more systematically into the analysis, whilst in the long term, upgrading the system to obtain continuous records would be the best solution.
Similarly, the observed increase in the number of rockfalls during spring (Fig. 7, Fig. A13) may result from a misinterpretation of smaller snow avalanches in the vicinity of the network. Accessing monitoring cameras would help increasing the knowledge of the seismic signature of these different surface phenomena.

Another drawback of working with triggered data is that several detections may be contained in a single file. The CNN
AlexNet configuration does not allow to use differently sized spectrogram images. In order to circumvent the limitation, we mask the signal parts in the spectrogram images that are not related to the detection (Fig. A17b). However, many of these events are not well classified, most of them are declared as regional events. This behaviour may also explain the unequal distribution of the number of regional events throughout the year. However, files with multiple detections are relatively seldom, such that the interpretation of the classified events should not be overly affected.

The dataset is not limited to the classes defined in this study and actually contains an even richer diversity of signals (Christiansen, 2021). In particular, complex slopequakes with precursors and hybrid slopequakes as defined by Provost et al. (2018) were found as well. An example of such signals is given in Fig. 12. This example illustrates the complexity of the signals recorded at Åknes and the problem of adapting the classification approach to the case under consideration. The automatic
classifier defined the event as a LF slopequake. However, if the computation parameters for the spectrograms are changed to enhance the resolution (here, a sliding window of 0.25 s with a 90% overlap), a weak, but clear phase emerges before the main burst of energy. On the broadband seismometer (Fig. 12b), a frequency decay of the main peak is visible from approximately 20-30 Hz to 10 Hz after 2.5 s, as would be expected for an hybrid slopequake. This behaviour is less obvious on the geophone (Fig. 12a). In order to explore more classes, for example the spectrogram computation would have to be reduced to the event
duration, resulting in different sizes of the input images, which is not possible if using a pre-trained network such as AlexNet.



Earth **Surface**
Dynamics
Discussions

EGU



**Figure 11.** Vertical component records and spectrograms recorded on the broadband seismometer (AKN) showing (a) a M2.7 earthquake located in the North Sea, and (b) an avalanche in late spring 2016 which destroyed the cable to geophone 2. Data are band-pass filtered in the range 2-80 Hz.

Earth **Surface**
**Dynamics**
Discussions

EGU

**Figure 12.** Example of a complex slopequake with precursory signal as defined by Provost et al. (2018). The level of detail necessary for its classification requires higher resolution computation parameters and is difficult to capture by the CNN. (a) Signal recorded on the North component of (a) a surface geophone, (b) the broadband seismometer (AKN). The white vertical dashed lines mark two phases, the second being the most energetic.





Moreover, events that are recorded simultaneously on the geophone network and the broadband seismometer installed on the central slope are rare.

Locating the microseismic events could help to better constrain the classification and vice-versa. Nevertheless, event location
remains difficult due to the extremely heterogeneous and fractured subsurface material at Åknes. Active seismic profiles were acquired during various field campaigns in summer (Tassis and Rønning, 2019) and show strong velocity contrasts, especially in the backscarp area ranging from less than 2 km/s to more than 5 km/s in only 50 m of material. Several attempts were made to locate the events. Fischer et al. (2020) employed a migration-stacking approach and a homogeneous velocity model with a P-wave velocity $V_P$ of 1500 m/s, fixing the event depth to the surface. Especially, the use of a homogeneous velocity
model may be inappropriate. More recent attempts using a 3D velocity model did not yield event locations within reasonable uncertainty limits, i.e. not more than a few tens of meters (Dahl, 2020). Since depending on the study, the sliding plane depth has been estimated between 50 and 120 m (e.g. Kveldsvik, 2008; Nordvik et al., 2009), a sufficient depth resolution of event locations is required. In addition, locating the events is hampered by the proper identification of the phases in the seismograms as illustrated in Fig. 12. Inverting the proper observations with the correct associated velocity model is fundamental whatever
type of location algorithm would be used (e.g. picking or migration-stacking approach).

Finally, although the classifier delivers satisfying results (Fig. 5), it still can be improved. First of all, the spectrogram stacking approach is not optimal, especially for signals exhibiting a strong variability in their waveforms for different geophones (e.g. Fig. A4 and A5) such as rockfalls. One way of tackling the problem would be to classify each spectrogram separately.
The chosen class could be the predominant one or the one which returns the largest combined probability. In addition, such an approach would enable to significantly increase the size of the training set, which could ultimately result in even better defined classes, taking into account all the existing waveform variability. Secondly, events related to surface processes such as rockfalls and snow avalanches lack ground truth. This could be partially resolved if web cameras were available. Even though it is not possible to monitor the whole slope, only a few exemplary observations of events of each class would be sufficient to identify
the related signal characteristics.

## 6 Conclusions

In this paper, we characterised and automatically classified seismic signals recorded by an eight-geophone surface network on the Åknes unstable rockslope. Five classes of events directly related to the slope could be identified following the typology proposed by Provost et al. (2018), including high-frequency (HF), low-frequency (LF) and successive slopequakes, short tremors
and rockfalls. Fifteen years of data, corresponding to approximately 60,000 events, were classified and analysed.

Similarly to the observations by Fischer et al. (2020), the seismic activity displays a strong seasonality with the majority of events occurring in spring, corresponding to both the period of snowmelt and temperatures oscillating around the freezing



point. This observation is in agreement with displacement rates of the Åknes slope which feature seasonal variations as well,
i.e. acceleration in spring and autumn (Grøneng et al., 2011).

This periodically increased seismic activity is accompanied by a significant increase in the number of detected low frequency
slopequakes, tremors and rockfalls. Comparatively, the amount of high frequency and successive slopequakes seems to remain
almost stable throughout the year. Our hypothesis is that HF and successive slopequakes are related to shearing on the sliding
plane. LF slopequakes and tremors, on the other hand, are associated with the thawing-freezing process within small cracks.
Since tremors are more frequent during daytime, they could be related to water movement within newly created cracks after
thawing, while LF slopequakes could correspond to the backscarp opening and the formation of new cracks. It is difficult to
distinguish if the apparent increase in rockfall activity in spring is realistic or if it is caused by other surface processes such as
local snow flows. Their increase in beginning of autumn coincides with increased precipitation. Finally, larger events such as
snow avalanches and granular flows may occur but are more difficult to characterise without continuous data and ground-truth.


No correlation of the seismic activity with precipitation could be established in this work. For example, strong rain episodes
(>100 mm within a day) are not necessarily followed by an increase in the microseismic activity. This does not mean that
such a correlation does not exist, since for example increases in the groundwater level are correlated with higher displacement
rates (Nordvik and Nyrnes, 2009). However, recent work aiming to better understand the hydrogeological system at Åknes
emphasized its complexity and showed the existence of barriers maintaining a high groundwater level at high elevation (e.g.
Sena and Braathen, 2021; Pless et al., 2021). Establishing potential systematic correlations between classified microseismicity,
precipitation and groundwater level fluctuations should be the subject of future work.

In the future, we plan to take advantage of the seismic data recorded by the broadband seismometer and the borehole geo-
phones that have been installed on the slope, but are not yet part of the monitoring system. Since these instruments record data
in a continuous mode, they may help providing more exhaustive catalogues of microseismicity. Last, but not least, the classifier
implemented in this study achieves a success rate of up to 80% . This could undoubtedly be improved further by applying more
advanced techniques within the rapidly flourishing field of machine learning.

*Data availability.* The broadband seismometer (AKN) data are part of the NORSAR station network (**NO**; DOI: 10.21348/d.no.0001) and
accessible through standard FDSN and EIDA web services. Geophone data will be made available during the review process.





# Appendix A: Classification

## A1   Diversity of seismic signals

Figures A1 to A8 show the vertical component records (left) and associated spectrograms (right) of the example signals plotted
in Fig. 2 to illustrate either the relative similarity or the variability of the waveforms across the geophone network.

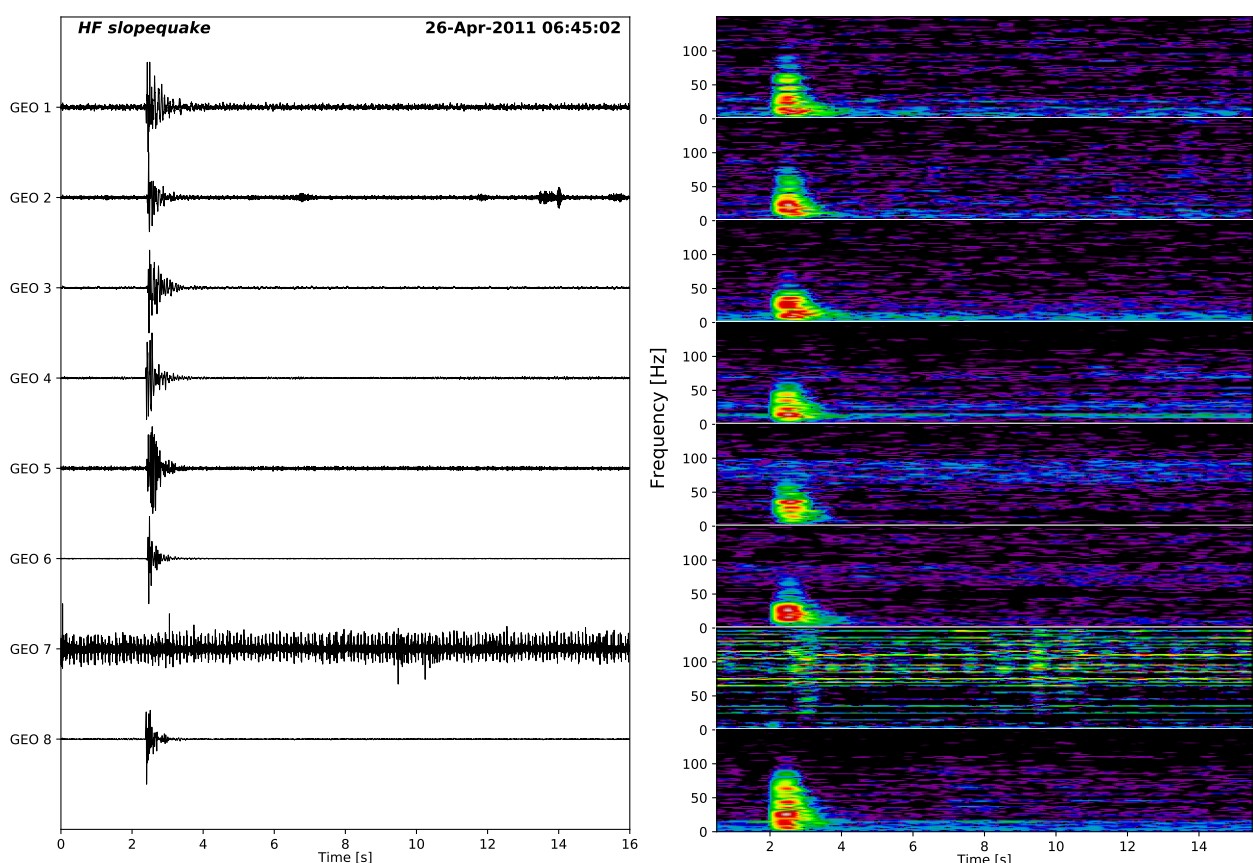

**Figure A1.** Example of a high frequency (HF) slopequake. Geophone 7 was not functional.

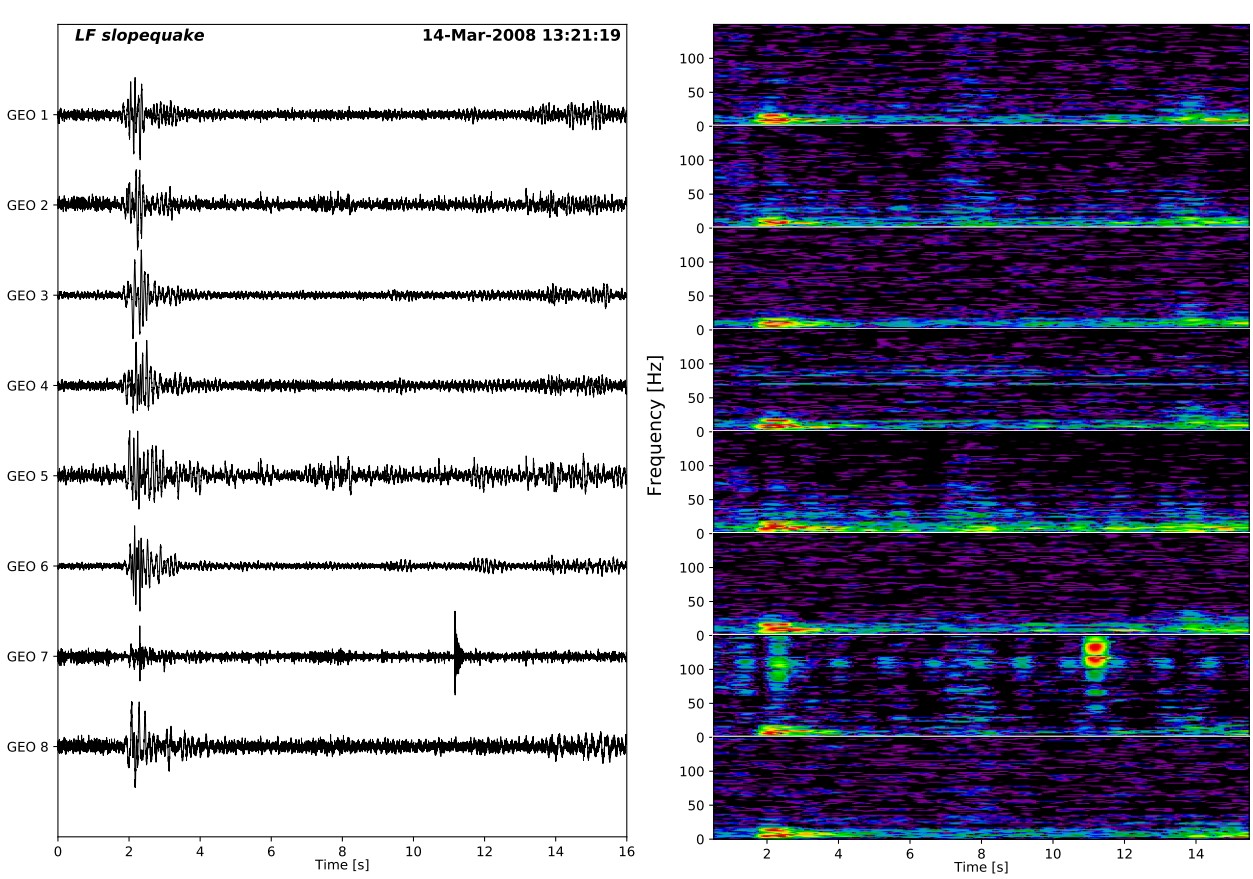

**Figure A2.** Example of a low frequency slopequake. A stronger spurious signal was recorded on geophone 7.





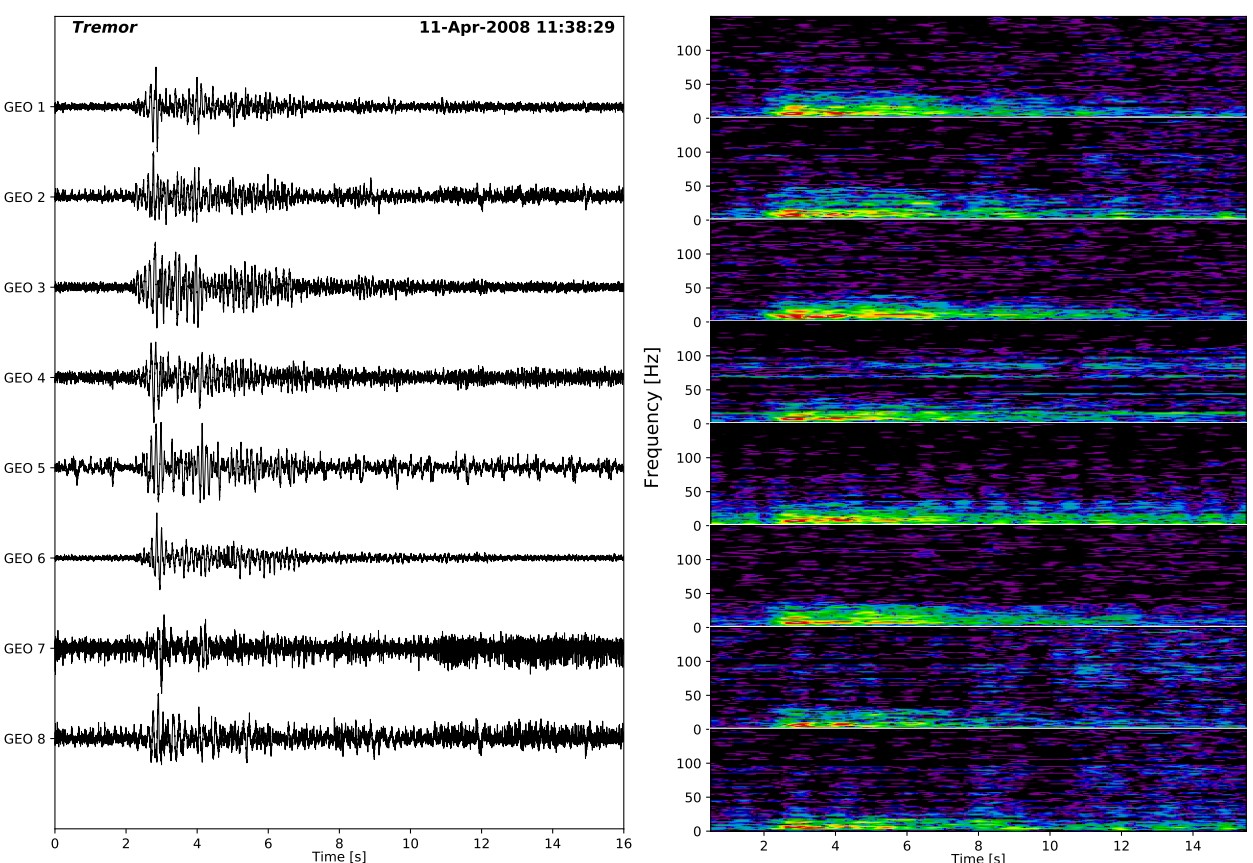

**Figure A3.** Example of a tremor-like slopequake.





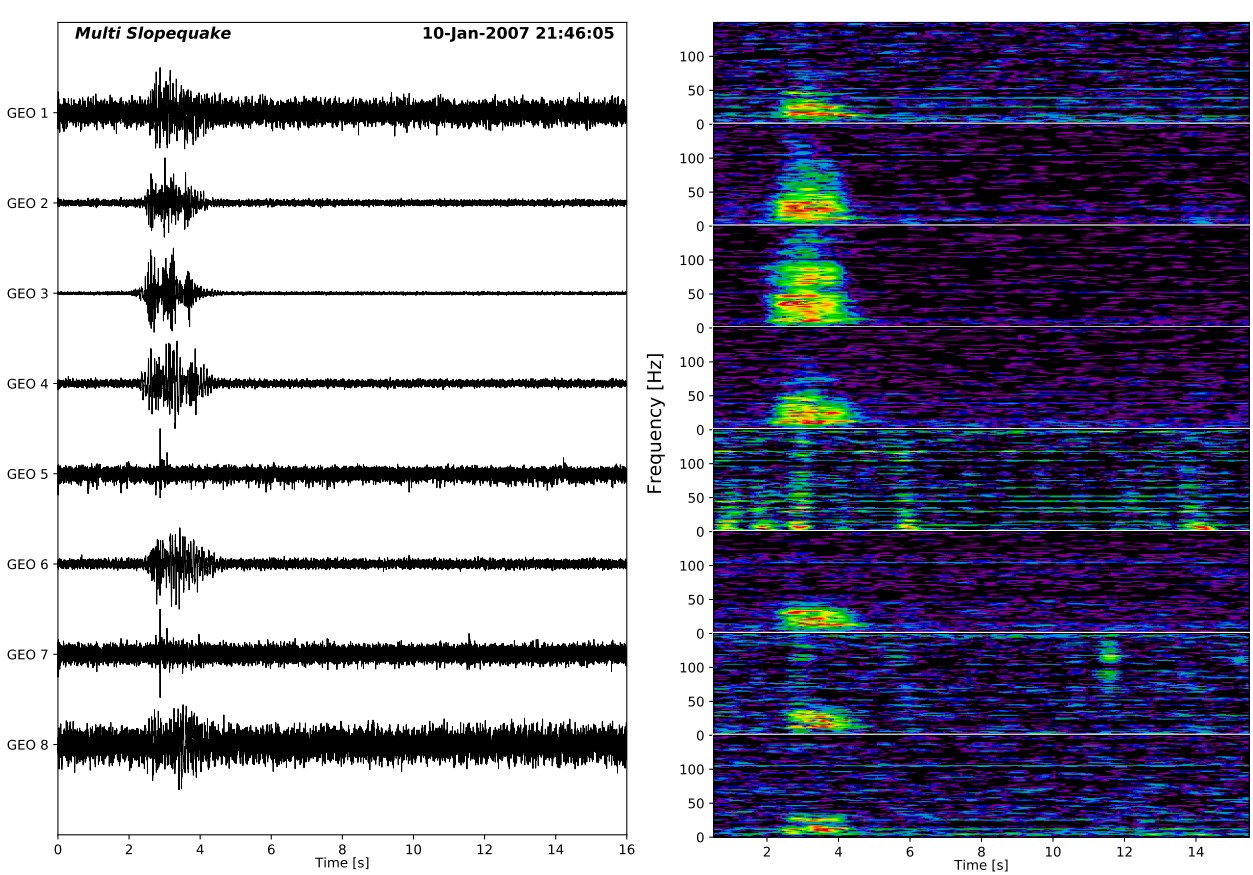

**Figure A4.** Example of a succession of slopequakes. Geophone 5 was not functional while geophones 1, 7 and 8 were noisy.

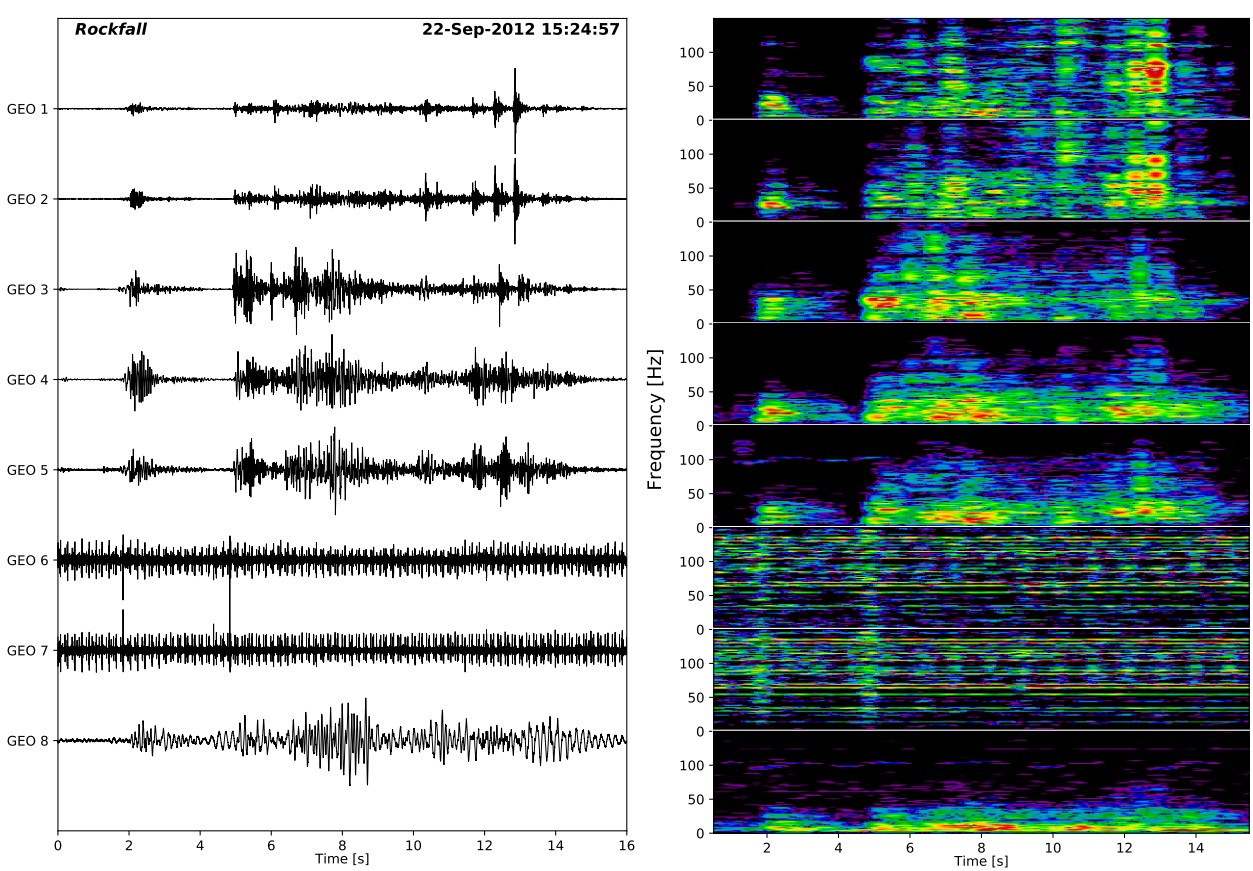

**Figure A5.** Example of a rockfall. Note the distinctive variability of the waveforms across the geophones. Geophones 6 and 7 were not functional.





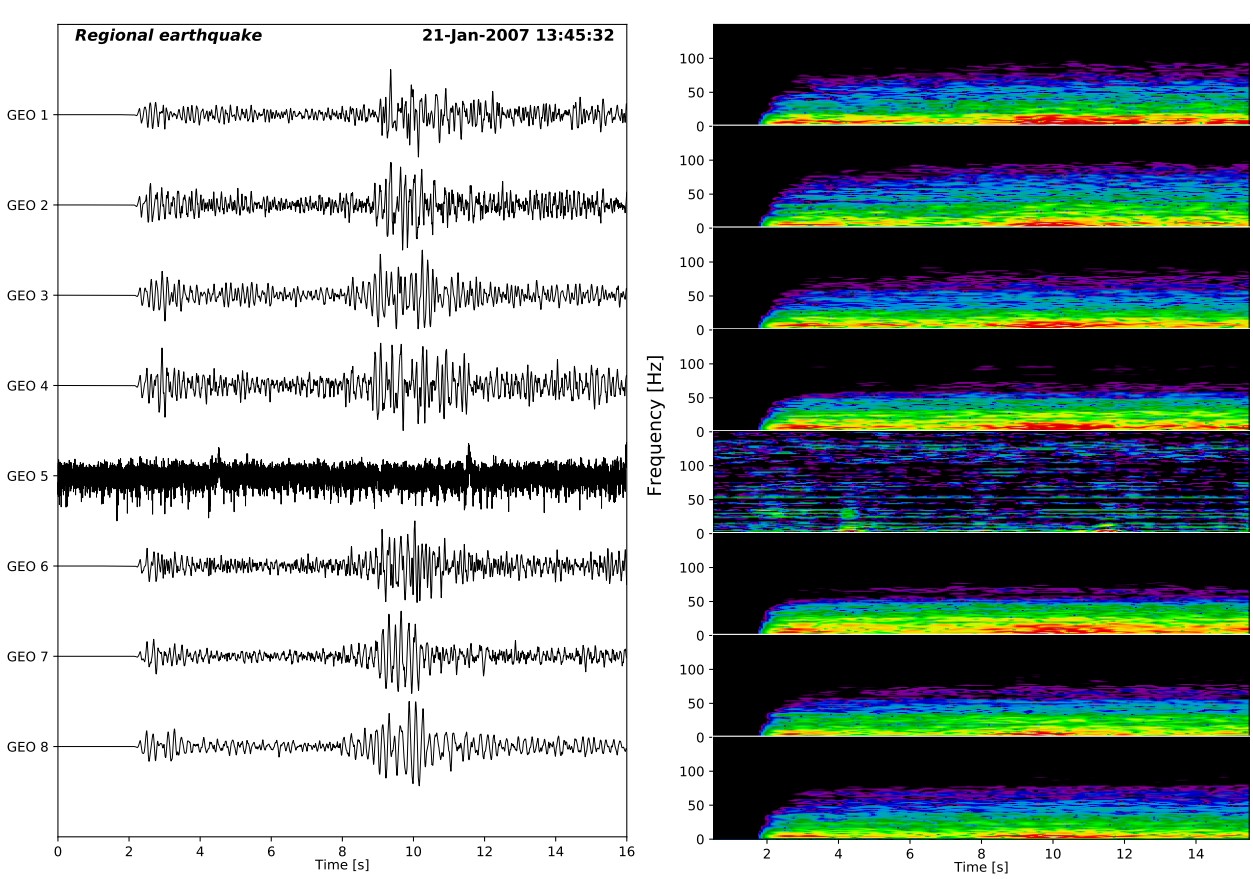

**Figure A6.** Example of a regional earthquake. P- and S-wave arrivals are clearly identifiable. Geophone 5 was not functional.



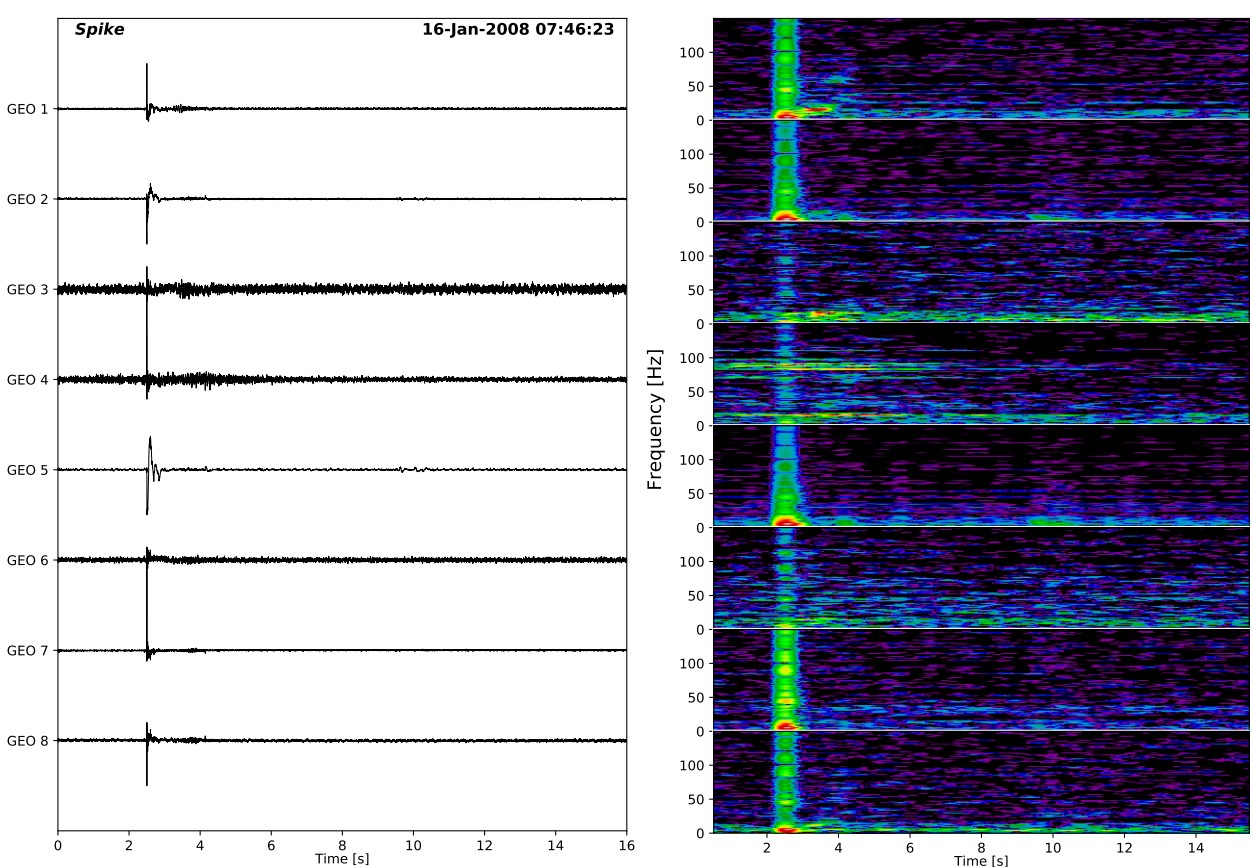

**Figure A7.** Example of an electronic spike.



Earth **Surface**
**Dynamics**
Discussions



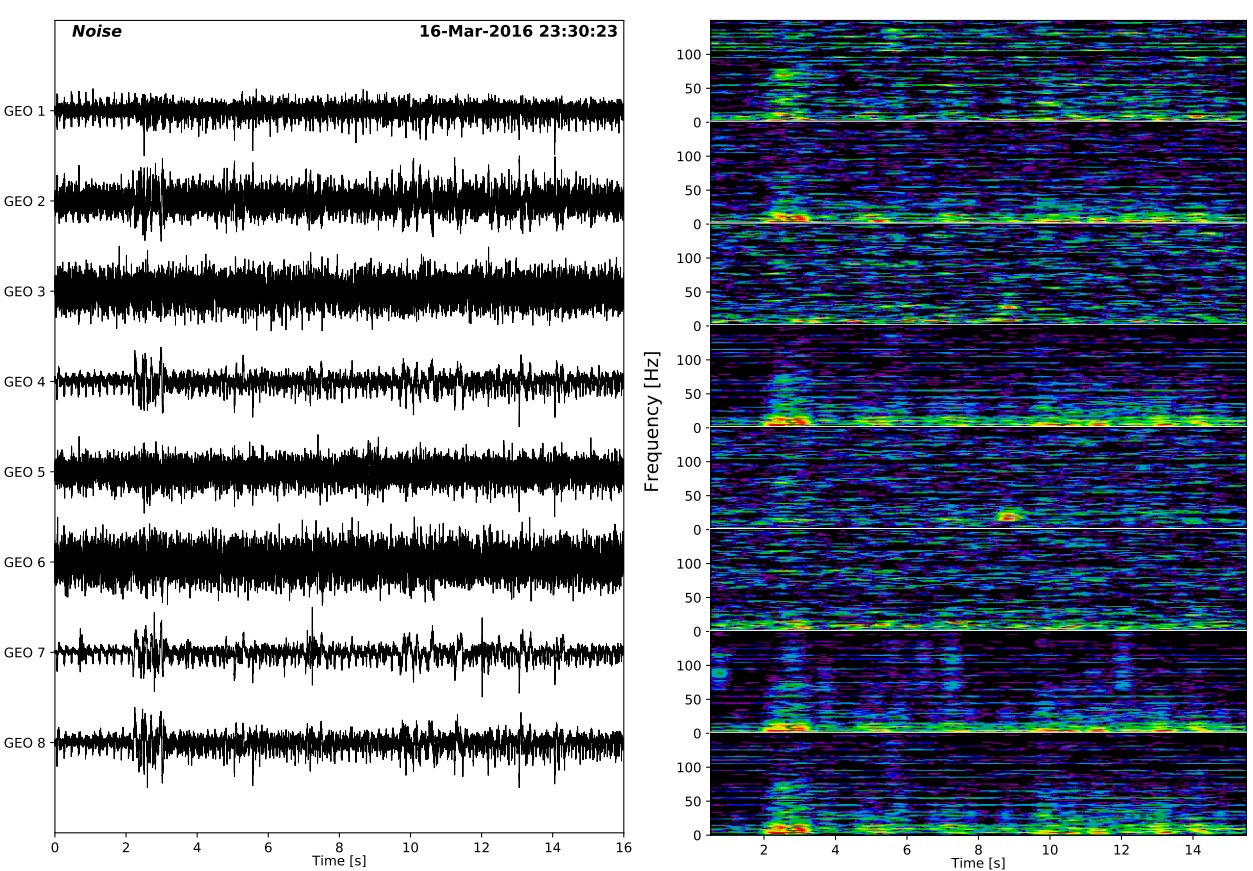

**Figure A8.** Example of a noise record.

Earth **Surface**
**Dynamics** Open Access
Discussions

EGU

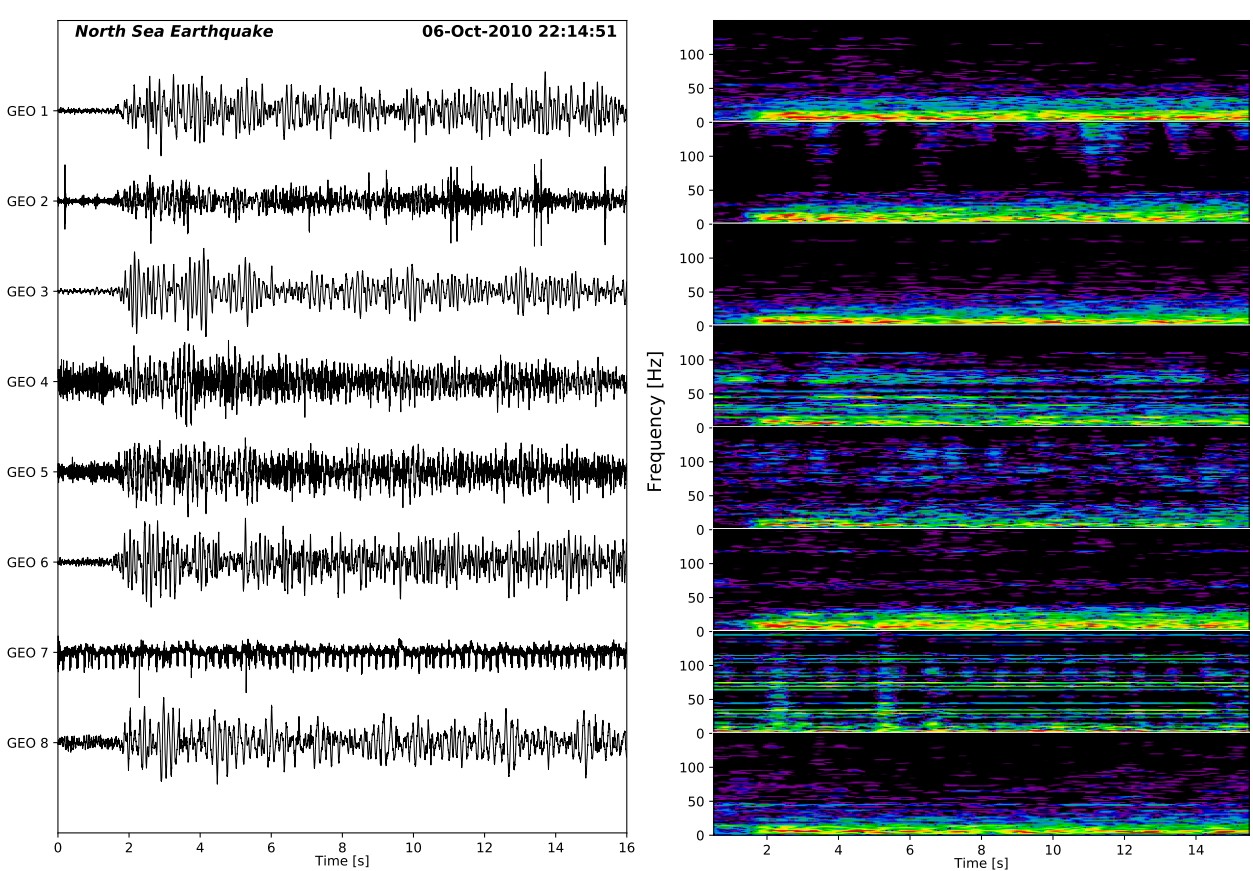

**Figure A9.** Example of a regional earthquake of magnitude M2.7 located in the North Sea, SW from Åknes. Due to the distance of the hypocentre to the stations, only the P-wave arrival is visible on the 16 s-long records. Waveforms recorded on geophones 2 and 4 are noisy and show greater variability compared to the other waveforms. Geophone 7 was not functional.



Earth **Surface**
**Dynamics** Open Access
Discussions
EGU

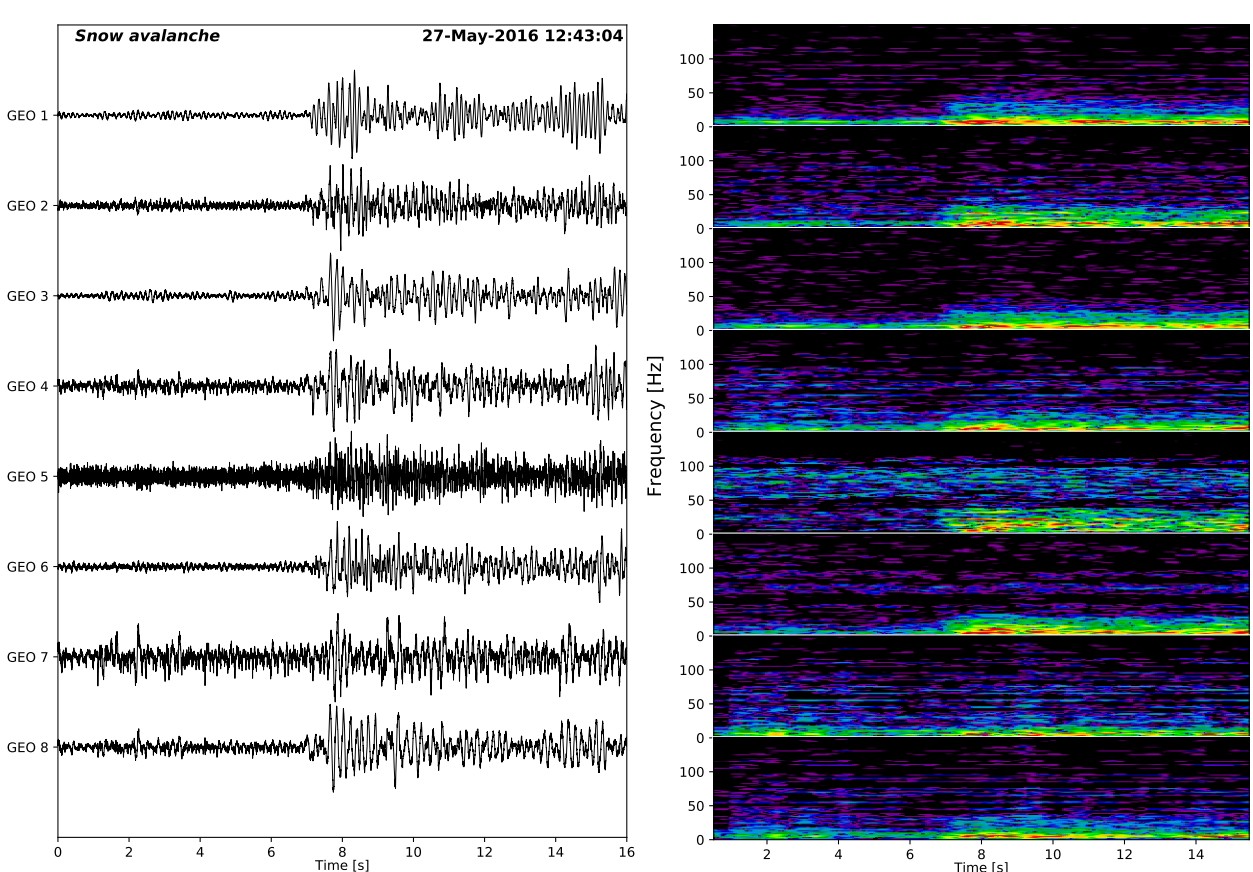

**Figure A10.** Vertical component traces of the event shown in Fig. 11. A first, weak arrival is visible around 2 s before a second, stronger arrival at 8 s. These observations are misleading and could be wrongly interpreted as P- and S-wave arrivals, respectively.



## A2    Classifier

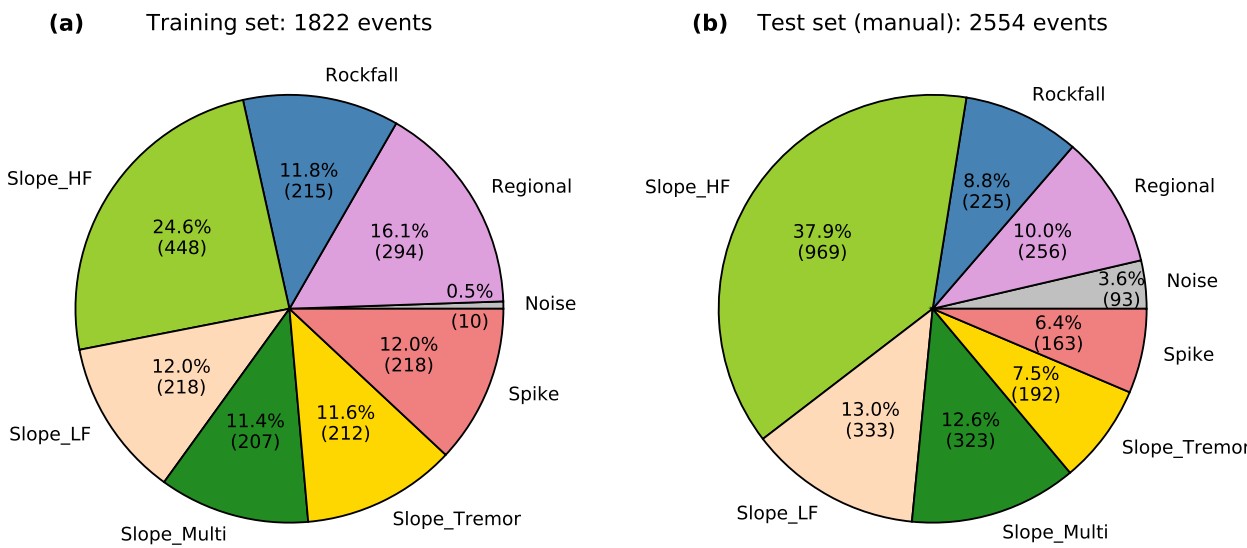

**Figure A11.** Pie diagrams showing the distribution of events within the different classes (a) in the training set and (b) in the test set, both expressed in terms of number of events and percentages.

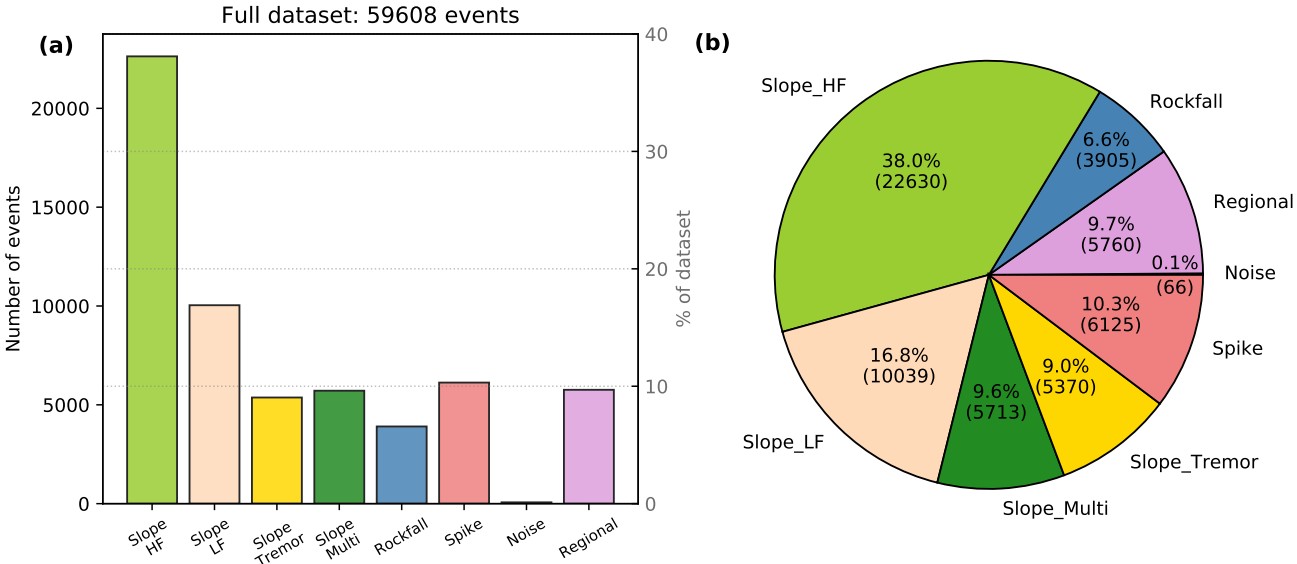

**Figure A12.** Distribution of automatic classes over a time period of 15 years (2007-2021); (a) histogram, (b) pie diagram.



Earth **Surface**
**Dynamics**
Discussions

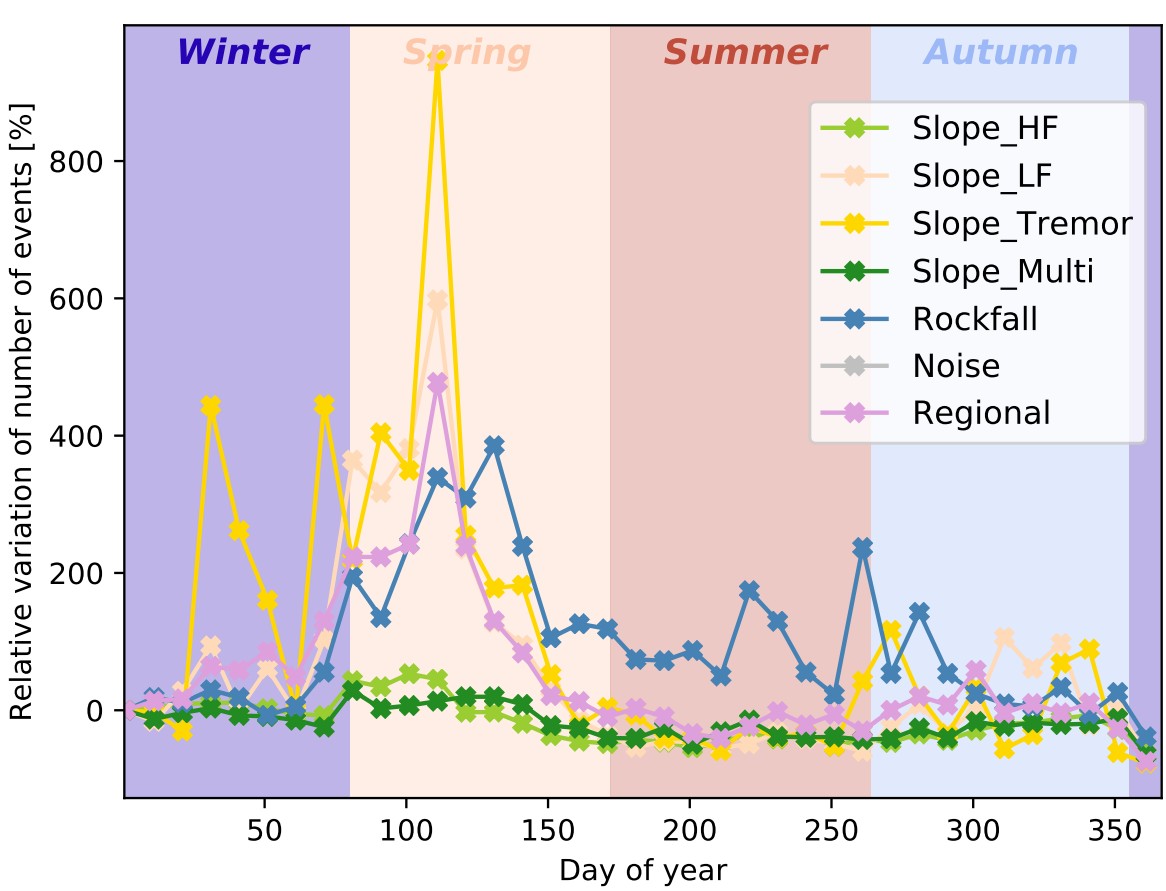

**Figure A13.** Relative variations $R$ of the number of events $N$ per class over chunks of 10 days. The reference value is taken from the first chunk, i.e. we compute $R = (N - N(1))/N(1)$. Note that noise events are so rare that they are not visible in the plot.





**Figure A14.** Box plots showing the distribution of amplitudes for each event class by season. Amplitudes in summertime are systematically higher than during the rest of the year.



Earth **Surface**
**Dynamics**
Discussions



**Figure A15.** Histogram showing the distribution of events of lower amplitude (light grey), intermediate amplitude (grey) and higher amplitude (dark grey) over months. The number of events is normalised in (a) by the total number of events in the different amplitude ranges, in (b) by the total number of events per month and in (c) by the total number of events in the dataset. The deficit of low amplitude events from June to October is particularly visible in all cases. A seasonal variation in the proportion of intermediate and high amplitude events exists as well, but in a much lesser extent.





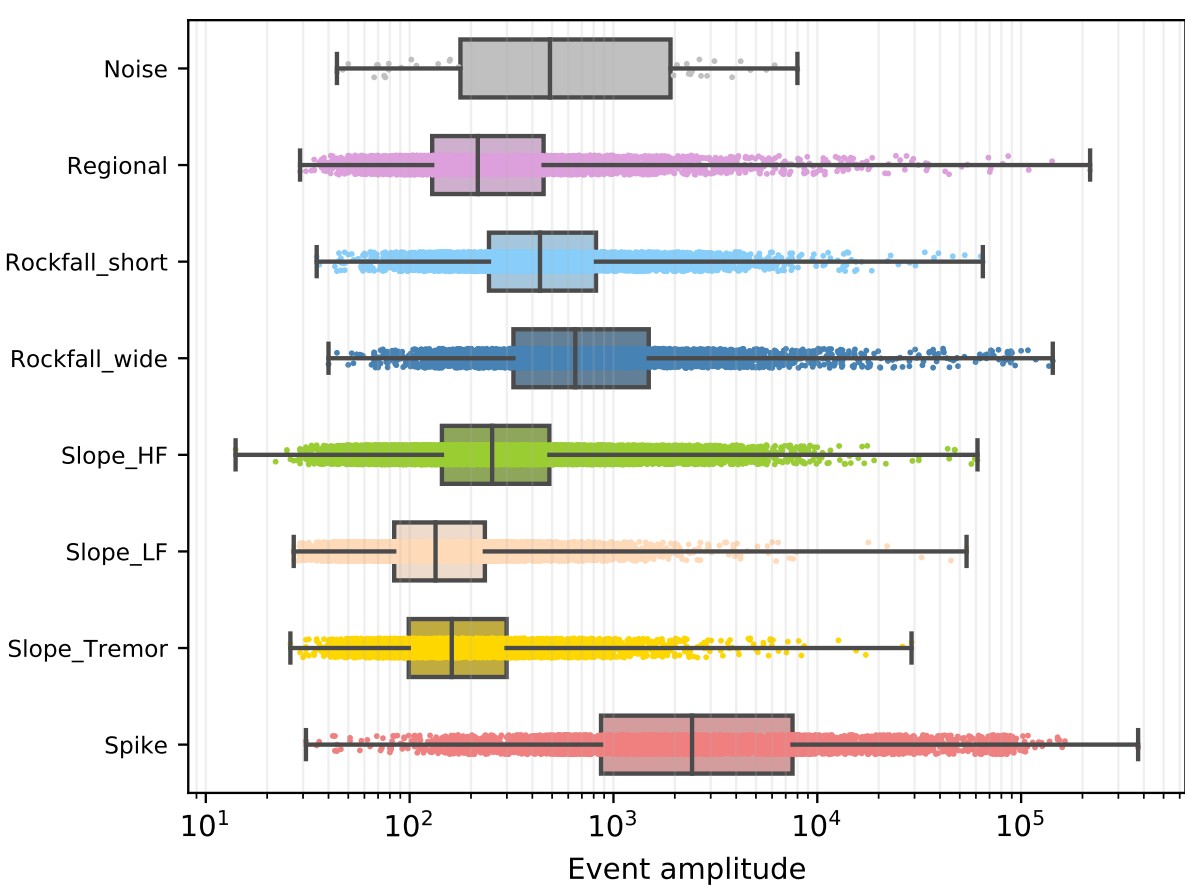

**Figure A16.** Box plots showing the distribution of amplitudes for each event class.



Earth **Surface**
**Dynamics**
Discussions

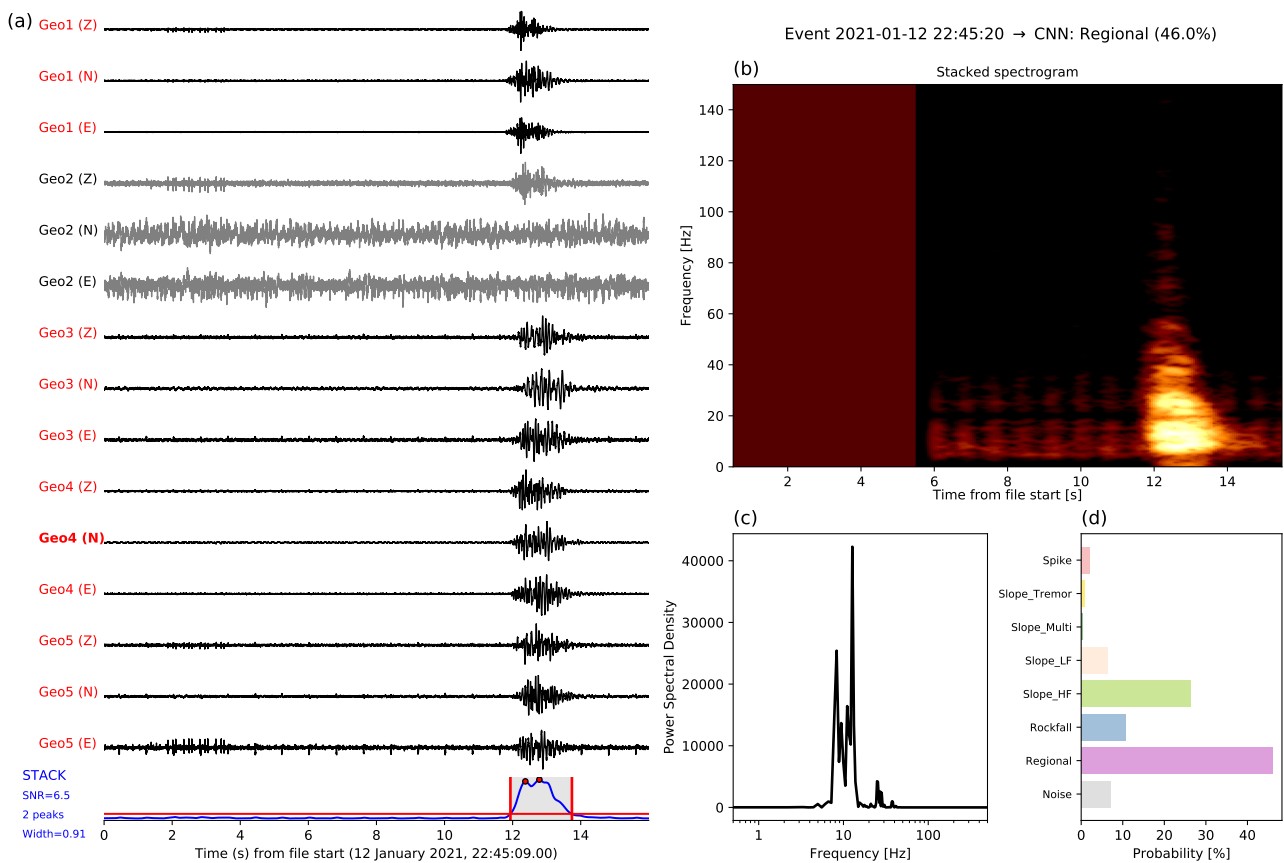

**Figure A17.** Example of an event wrongly classified as regional earthquake instead of HF slopequake, see Fig. 4 for description. Note that we only display the waveforms at geophones 1 to 5, since geophones 6-8 were not functional. The event file contains two events and detected electronic noise at around 2 s. A mask was applied on the spectrogram image, but cannot prevent misinterpretation. However, (d) illustrates that the probability to be a regional event is relatively low (below 50%) and the probability of being a HF slopequake is not much lower.

Earth **Surface**
**Dynamics**
Discussions



**Figure A18.** Probability density functions of some extracted features from the training set. Although the CNN is not a feature-based algorithm, it is interesting to look at the distribution of those features since they are the ones that a human eye would subconsciously catch and analyse while manually labelling the data.

Earth **Surface**
**Dynamics**
Discussions



## Appendix B: Meteorological data

The meteorological station is located in the vicinity of the upper bunker (Ørnereiret) close to the top scarp. Temperature and precipitation are measured and averaged hourly.

**Figure B1.** Temperature variations recorded at Åknes. The measurement period spans over 15 years (Jan. 2007-Dec. 2021). (a) Hourly variations of average temperature for each month of the year. To obtain the relative variations shown in (b), the curves in (a) were normalised by their minimum and maximum values: $\Delta T/\Delta T_{max} = (T-T_{min})/(T_{max}-T_{min})$.

Earth **Surface**
**Dynamics**
Discussions

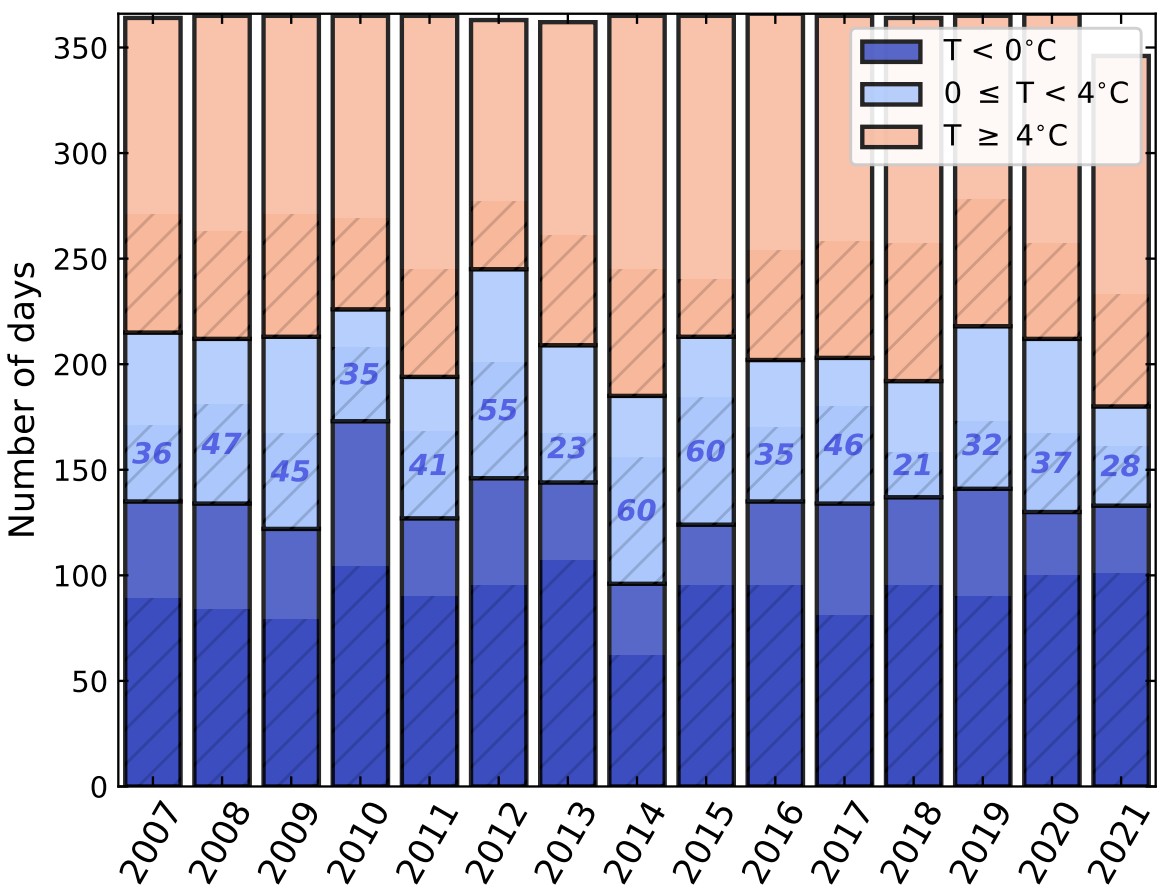

**Figure B2.** Distribution of the number of days per year when the average temperature is lower than 0°C (dark blue), between 0 and 4°C (light blue) or above 4°C (light red). Hatches correspond to days in the first half of the year (1st of January to 30th of June). The number of days when the average temperature is between 0 and 4°C during spring is indicated. Days with outliers (abnormally small or high temperature values) are removed.



*Author contributions.* Manual classification of events for the training set was performed by NL and FMJS. FMJS created the main workflow for the CNN and tested different algorithms and parameters. NL made additional tests, QC and implemented the classifier in the near real-time workflow. NL analysed and interpreted the results and wrote the paper.

*Competing interests.* The authors declare that they have no conflict of interest.

*Acknowledgements.* We thank A.M. Dichiarante (NORSAR) for creating Fig. 3. We are very grateful to D. Kühn and V. Oye (NORSAR)
for useful comments which helped improve the manuscript. We thank NVE for funding the seismic network, data acquisition, for facilitating fieldwork and for providing the meteorological data.



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
