# Peer review of "Automated classification of seismic signals recorded on the Åknes rockslope, Western Norway, using a Convolutional Neural Network"

_Earth Surface Dynamics, 2022_

## Referee Comment (RC2)

**Review comments**

Esurf-2022-15-manuscript

[Automated classification of seismic signals recorded on the Åknes rockslope, Western Norway, using a Convolutional Neural Network]

June 19, 2022

The manuscript presents a deep-learning-based classification model to automatically detect the microseismic events occurred at slope. Precision in the occurrence time of microseismic catalogue would be helpful to explore the temporal link between microseismic activity and temperature variations. I think that the subject is relevant to publication in Earth Surface Dynamics. However, there are several places where I think a bit more explanation and major revision is needed. Detailed comments are listed below.

*(1) Lines 92-98*

Introduction of seismic monitoring needs to be enriched with the literature review.

**Continuously seismic monitoring**: Chang et al. (2021), Locating rock slope failures along highways and understanding their physical processes using seismic signals, *Earth Surf. Dynam.*, 9, 505-517. They presented the physical processes of surface mass-wasting events using a series of the spectrogram features, which were comprehensively supported by the video records, eyewitnesses and field investigation.

In this manuscript, on the validation, I think the authors either forgot to or did not accurately present the microseismic event validation. The authors do not have direct evidences to support detected events (47,561 after regional earthquakes, spikes and noise are removed) linked to the rockslope site.

**Seismic precursors**: Schöpa et al. (2018), Dynamics of the Askja caldera July 2014 landslide, Iceland, form seismic signal analysis: precursor, motion and aftermath, *Earth Surf. Dynam.*, 6, 467-485. They presented the precursory seismic signals, source mechanism of landsling, and location of afterslide sequence. A series of numerical modeling was conducted to have a better understanding in mechanism of generation of precursory seismic signals.

In the current manuscript, actually, there are a lot of statements that are not certainly supported by the results, which are often possible, but they are not open-discussed based on the referred references.

**Seismic interferometry**: Kang et al. (2021), Rigidity strengthening of landslide materials measured by seismic interferometry, *Remote Sens.*, 13, 2834. They investigated the temporal link between daily relative velocity changes and in-situ measurements of slope to understand the possible rigidity strengthening.

In this manuscript, a broadband seismometer was deployed in 2009, which is available for analysis of seismic interferometry. Then, tiny velocity variations can be extracted to provide additional information in understanding the physical processes of slopequakes. More discussion should be added in the manuscript.

*(2) Line 107*
For the detector of the short-time-average/long-time-average STA/LTA trigger, successful capturing of seismic events depends on proper settings of the trigger parameters: (a) the average values of the absolute amplitude of signal in two consecutive moving-time windows; (b) a pre-set value of the STA/LTA ratio. I cannot find aforementioned parameter settings in the manuscript.

*(3) Lines 132-139*
There are two methods for constraining S-wave velocity structure at slope scale:
(a) Rayleigh-wave ellipticity from polarization analysis of seismic ambient noise (e.g., H/V spectral ratio).
(b) Dispersion curve inferred from the seismic ambient noise cross/auto-correlation functions.

Thus, a combination of P-wave (seismic reflection profile) and S-wave velocity model is helpful to location source of detected signals. The velocity model is not always difficult to establish.

The second question is related to identification of seismic phases generated by the microseismic events. In fact, it is very difficult to distinguish the P- and S-wave due to the complexity of the source mechanism and the dominance of surface waves in the slopequake-generated seismic signal. However, analysis of three-component seismogram can help us to identify the types of seismic phase, such as the particle motion analysis.

A possible solution for aforementioned problems is the amplitude source location (ASL) method (Chang et al., 2021), which can estimate the source location by using the large bursts of seismic amplitude without a priori knowledge of velocity model

and seismic phase. Such methods, the reader may want to find explored and clarified by the authors.

*(4) Lines 141-146*

At least, the preliminary location of microseismic event is required to support the manually labeling (slopequakes, tremor, rockfall) used in this study. Otherwise, you cannot study the seasonality pattern of microseismic activity, also resulting poor understanding their source mechanisms (origin of slopequake and tremor). Above questions are also existed in the previous study of Provost et al. (2018). You need have some advances compared to Provost et al. (2018).

*(5) Lines 166-169*

In fact, the motion behavior of rockfall is composed of falling, sliding, rolling, and bouncing, which are dominated by the slope angle except for the sliding. For example, as the slope angle is less than 45°, rock mass tends to roll.

*(6) Lines 177-179, 294-296*

All records are extracted by using the STA/LTA detector, thus, you cannot have the noise label. As the example of noise signal and spectrograms shown in Fig. A8, there are an obvious and emergent signals with a short duration (the time point from 2 to 4 seconds). I am sure that is not noise signal. And, that's the reason why the noise label can be easily classified as the source types of regional earthquake (32.3%) and tremor (10.8%) based on the current model (see Figure 5). I am completely not surprised to see above result.

*(7) Lines 214-227*

Actually, seismic features observed in this manuscript (such as signal duration, dominant frequency content; see Table 1) can be easily be automatized by machine-learning-based classification/picker/detector approaches.

*(8) Lines 245-251*

For a specific source (events associated with the slope or surrounding area; Figures A1-A8), there are the clear discrepancies in signal arrival-time, frequency content, and spectral amplitude, which would be useful to constrain the possible source location, further understanding the origin of source mechanisms of microseismic events. I suggest that only stacking the three-components seismograms for each station and building the single station-based training model to detect the seismic signal individually. In practical real-time application, once the number of triggered

station reach a certain threshold (for example, larger than three station), location approach can be implemented.

**(9) Line 250** "remove low-quality traces based on a user-defined threshold"
Provide a certain definition of a user-defined threshold used in this study.

**(10) Figure 4**
The overall flavor of argumentation is somehow "positive-example"-based, rather than generic and equally balanced. In several cases I can read a lot about the positive outcomes and do not read negative cases. One example is the statement in the section of "Near real-time implementation" and Figure 4.

**(11) Line 309**
The authors do not have direct evidence to support detected events in this study linked to the rockslope. Current manuscript did not convince me, even though I can feel the enthusiasm of the authors.

**(12) Line 317-320**
Please make a systematic comparison between detected catalogue in this study and dataset reported by Fischer et al. (2020). Such comparison, the reader may want to find explored and clarified by the authors.

**(13) Figure 6**
(a) Please add time-series of rainfall, groundwater level, and borehole displacement
(b) What's the range of magnitude of regional earthquake? Can the authors predict the peak ground-motion amplitude by using the empirical ground-motion prediction equation? If YES, you can see what's happened in relation between slopequake activity and ground shacking.

**(14) Lines 325-329**
Seasonality pattern can also be influenced by the detection capacity of network. Snow melting and precipitation can increase the noise level due to the rainfall water dropping to ground and high flow dynamics in river channel and gully at slope (water-generated seismic noise). Please provide more detail discussion about this question. Above statement also supports that the number of most of seismic sources show decreasing trend during the summer months (June to September).

*(15) Lines 329-337*

Can the authors estimate the changes of forces/pressures caused by the variations in water density? Then, how large magnitude of slopequake can be triggered due to aforementioned state changes?

*(16) Line 358-361*

This is an unfair estimation. The mean of maximum amplitudes is significantly depended on both of the frequency and distance of wave propagation. Thus, current maximum amplitude used in this study (Figure A14) can be directly used to represent the event size.

*(17) Lines 388-397*

If an additional broadband seismic station far away from the slope sit was included, it is very easy to identify the regional earthquake event or microseismic events on surrounding hillslope.

*(18) Figure A7*

What is an electronic spike? If it is due to the instrumentation problem (data logger), why spike signals can be observed at same time for all stations?

---

## Author Response (AR1)

**Reply to reviewer #1**

Dear Editors, dear Reviewer,

We thank the reviewer for reviewing our manuscript. We went through all comments carefully and replied to each of them in the following. The reviewer's comments are listed and repeated in italic. Clarifications and improvements suggested by the reviewer that we took into account in the revised manuscript are highlighted in **bold green**.

"This paper is very interesting because it proposes for the first time to use Convolutional Neural Network (CNN) on seismic signals recorded on unstable slopes. The analysis of the catalog produced is very exhaustive, relevant and brings very interesting elements to better understand the link between the micro-seismicity endogenous to a landslide and the external forcings, and the associated mechanisms. The article is very well written, easy to follow and understand, and the figures are of excellent quality."

"Overall, I find this article almost publishable as is. I nevertheless have some minor comments and questions, especially on the Machine Learning part. I detail my comments below:"

- "L202-205: I think the training set should be better described. In any implementation of supervised classification algorithm it is very important to know the exact number of events used to train and to test the algorithm, as those can greatly influence the performance of the algorithm. Then the description here is confusing: first you say that "most classes constitute around 12% of the training set", but then that this is not the case for the HF SQ and RE classes. I would suggest adding two columns in table 1 with the number of events in the training and in the testing sets for each class." → we agree that this may be confusing. The composition of the training and test sets is shown in the form of pie diagrams in the appendix only (Figure A11) and numbers are provided there, but following the suggestion, we have now added two new columns to Table 1.
- "L216-219: Some previous studies also proposed features computed on the spectrograms, and they usually are amongst the best features for the classification (e.g. Provost et al., 2017; Hibert et al., 2017; 2019; Maggi et al., 2017; Wenner et al., 2021). This should be mentionned here. You propose a new approach based on the spectrogram but you are not the first to propose to use this data transformation and this should be acknowledged in your description here." → you are right, these papers are cited earlier in the manuscript but not specifically linked to the features definition/extraction part. In the revised manuscript, we added a new sentence (1. 229-230).
- "L222-227: I don't understand the drawbacks described here. Features computation is not more complicated than the computation you do to transform your data into spectrograms. I understand that you need some arguments to tell the readers why you choose to work with spectrograms, but I don't think the arguments you propose here are valid." →

we agree that computing features is not substantially more complicated. But they need to be defined beforehand, resulting in a large number of different features. Moreover, it certainly depends on the way features are extracted. In a previous approach that we tried, we found that the event duration was a major feature for discrimination – but it was also not the easiest feature to extract (in the sense that it is rather mistakeprone). Since we also extracted most features within the event duration window, if this was not well estimated, the other features usually contained more errors and could not be well estimated (e.g. the energy was over- (or under-) estimated,...). But we agree that this was the case for only a few data and that this cannot be generalised or used as a good argument "against" feature-based algorithms. **Therefore, in the revised manuscript, we removed some sentences and reformulated the whole** (around 1. 225).

- "One could argue that your approach based on spectrograms is more susceptible to computation parameters, as the resulting spectrogram is completely controlled by the way you compute it (e.g., what is the influence of the spectral transformation you use? Between DFT, wavelets, Z-Transform, etc? What about the window shape? the window length? the overlap?)." → We have not tested different transforms, although they might of course slightly influence the final results. Extracted features are dependent on computation parameters in a comparable way: for example, features extracted from spectrograms will still depend on the way spectrograms are computed and spectral features may also vary if the FFT or the PSD is used.
- "What would be the best option is the possibility to input directly the raw signal into the machine learning model, but I'm not sure you can do this with CNN."  $\rightarrow$  it is possible to use 3C data as input as done by e.g. Köhler et al., GJI, 2022 (Classification of seismic calving events in Svalbard). In the Åknes case, it has been briefly tested, but spectrograms seem to yield better results.
- "I think that there is a simple argument in favor of CNN that you should make, which is that supervised machine learning algorithms based on curated features might miss critical information within the signals that CNN will find because it does not need any manual hence subjective definition of the features. CNN use images (most of those models at least), so in order to use them on seismic data you need to transform the signal into something that has the same properties as an image, which are spectrograms. This is largely sufficient to motivate the use of CNN and your study I think." → Thank you indeed, this was also our point, but it was maybe not as clearly expressed. See aforementioned correction..
- "L249: You choose to stack the spectrograms, but how is this influencing the global noise of your final spectrograms? Would it be better to calculate the product of the different spectrograms? I might be wrong, but by multiplying the spectrograms I think that you will reduce the noise and the influence of propagation effects while bringing out the part of the seismic energy generated by the source? It might be worth testing in a future work." → the stacking approach indeed enables to enhance the signal-to-noise ratio. We did not think about multiplying the spectrograms instead of stacking them, but we thank you for this good idea as it should indeed enhance the SNR even more.

- "L274: For how many of those 59.608 events the class has been confirmed manually? The 2554 you included in the test set or more than this?" → Only the 2554 events constituting the test set have been QC'd manually.
- "If more, what guided your choice of the 2554 events you have in your test set?"  $\rightarrow$  The test set contains all detected events since November 2020 when the classifier was first implemented in the near real-time workflow. In addition, it contains randomly selected events that were also checked while testing the classifier before its implementation.
- "Are the events in the training set from those 2554 events?"  $\rightarrow$  No, all those events are different from the training set.
- "A better description of your training and testing data sets is needed I think, as suggested in a previous comment." → Columns with number of events in each class for both the training and test sets will be added in Table 1 as suggested in your previous comment.
- "L303-304: So you did scan all the 59.608 events manually to remove the electronic spikes? See comment above. It should be very clear for the readers if the catalogue you interpret in the following sections is fully automatically made, automatic but fully manually controlled, or automatic but partially manually controlled. Provide numbers." → Thank your for this comment. This part was indeed not well described in the manuscript. The catalogue was only adjusted for the cumulative energy curve in Figure 6, where all events not related directly to the slope movements were removed (i.e., spikes, noise, regionals). Using the automatic classes, we sometimes observed large jumps in this curve and checked which events caused them. We found that most of these jumps were due to spikes being wrongly classified, and we subsequently decided to remove them ( 450 events, i.e., 7% of the spikes were not well classified). However, in the rest of the paper (histograms, ...), we only used the uncorrected automatic catalogue (given the large number of events, those misclassified events count only for less than 1% of the dataset and do not affect significantly the results). We hopefully clarified this part in 1. 310-315 of the revised manuscript.
- "L400 & 414-415: Would it be possible to process data with different sizes with other CNN implementations?" → using CNN implies that input images have the same size, but FCN (Fully Convolutional Networks) allow for different sizes. Alternatively, one can transform the image to the desired size before feeding it into the CNN. Since we work with triggered data with fixed length, we did not feel a great need to explore more any of these options. We added a sentence, 1. 420 in the revised manuscript..
- "This is a huge advantage of methods based on curated features (RF, SVM), they can work with signals with different durations. However this needs a pre-detection of the event, which can be tricky. This is why we start to see implementations of those approaches on moving windows (Wenner et al., 2021; Chmiel et al., 2021). Would it be possible to do the same using a NN such as AlexNet? If so, it can be interesting to tell the readers in the discussion or in the perspectives how such an implementation could be done and what could be the difficulties." → Thank you for this question and the papers which are very interesting. Although we did not have the occasion to test our workflow on continuous data, we think this should be possible to apply a similar approach (i.e.

sliding window) with a CNN since the filters in the CNN can be seen as features. A work going into this direction has been published by Takahashi et al. (Earth, Planets and Space, 2021) with the aim of detecting and classifying earthquakes, tremors and noise using CNN.

Moreover, in a CNN, the detection step could be performed through class activation maps (CAMs) which help visualizing and highlighting which part(s) of the spectrogram image the CNN focuses on. It should then be possible to define activation thresholds above which an event would be first identified and then classified.

Lastly, in cases when several stations are available, one could also imagine classifying each trace separately and decide whether an event occurred by voting. For example, if more than 50% of the automatic classes at each individual traces are classified as noise, then there is no event.

We reformulated the last sentence of our conclusion (as suggested by your last comment) and added a few sentences on potential adaptation of the workflow to continuous data. We did not really add anything on related challenges, although expected (size of the training set,...). The choice of the length of the sliding time windows would also need to be adapted to the signals of interest, i.e. we may not be able to classify both local signals (slopequakes) and regional earthquakes, but since we are mainly interested in slopequakes, this should not be an issue.

- "L435 437: Indeed. This sounds like it could be easily tested. What prevented you from doing so for this study? Was it too costly in term of computation time?"  $\rightarrow$  we would like to draw your attention to a follow-up work that has been performed and is currently accessible on ArXiv: Lee et al., 2022, https://doi.org/10.48550/arXiv.2204.02697. In this work, the classifier has been improved by using ensemble prediction and self-supervised learning approaches. A direct comparison of classification performances for AlexNet when individual spectrograms are considered instead of stacked spectrograms is provided in the paper: unsurprisingly, the ensemble prediction approach turns out to be one of the major factors of improvement. As a side note, please note that this work has neither been applied to the entire Åknes dataset, nor in the near-real time workflow yet (although planned in the future). In the revised manuscript, we added the reference to Lee et al., 2022.
- "Conclusions: I found the last sentence/paragraph a bit vague and underwhelming. I think you have plenty of insights from this first implementation of a CNN that you should share with the readers. They should be highlighted in the conclusion."  $\rightarrow$  thank you for this comment, we agree that this sentence is too fuzzy. The conclusion was reformulated and previous comments addressed there.

Earth Surf. Dynam. Discuss., author comment AC2 https://doi.org/10.5194/esurf-2022-15-AC2, 2022 © Author(s) 2022. This work is distributed under the Creative Commons Attribution 4.0 License.

**Reply on RC2**

Nadège Langet and Fred Marcus John Silverberg

Author comment on "Automated classification of seismic signals recorded on the Åknes rockslope, Western Norway, using a Convolutional Neural Network" by Nadège Langet and Fred Marcus John Silverberg, Earth Surf. Dynam. Discuss., https://doi.org/10.5194/esurf-2022-15-AC2, 2022

Dear Editors, dear Reviewer,

We thank the reviewer for the many comments and some concerns about the manuscript. Most of the comments, however, seem to address the topics of microseismic event location and ambient seismic noise analysis, which both are only mentioned as supportive methods in our manuscript and are not the focus of our study, which is the classification of microseismic signals. Moreover, the raised concerns seem to be based on misunderstandings that we are trying to clear out in our reply below and in some clarifications in the updated manuscript text. The reviewer's comments are listed and repeated in italic.

"The manuscript presents a deep-learning-based classification model to **automatically detect** the microseismic events occurred at slope" --> This is a misunderstanding, the CNN is only used for event classification, not detection.

"Precision in the occurrence time of microseismic catalogue would be helpful to explore the temporal link between microseismic activity and temperature variations." --> We are not sure what is meant here, as we do investigate the link between temperature changes and microseismic activity. The microseismic catalogue that we use spans 15 full years, from 2007 to 2021 (§4.3). This information is repeated several times throughout the manuscript and is also contained in the captions of Figures 6 to 10. Seismic data are acquired with a 1000 Hz sampling rate. Temperature is measured with a 1-hour sampling rate and we also used records since 2007.

"I think that the subject is relevant to publication in Earth Surface Dynamics. However, there are several places where I think a bit more explanation and major revision is needed. Detailed comments are listed below." --> we provide detailed replies to these comments.

"(1) Lines 92-98

Introduction of seismic monitoring needs to be enriched with the literature review.

Continuously seismic monitoring: Chang et al. (2021), Locating rock slope failures along highways and understanding their physical processes using seismic signals, Earth Surf. Dynam., 9, 505-517. They presented the physical processes of surface mass-wasting events using a series of the spectrogram features, which were comprehensively supported by the video records, eyewitnesses and field investigation.

In this manuscript, on the validation, I think the authors either forgot to or did not accurately present the microseismic event validation. The authors do not have direct evidences to support detected events (47,561 after regional earthquakes, spikes and noise are removed) linked to the rockslope site." --> we are not sure what sort of evidence the reviewer would like to see: (1) the site is a known place of sliding and its instrumentation started 40 years ago. There is large and visible open fracture that is widening (and around which the seismic sensors are placed), as well as other geological evidence that are described in  $\S2.1$ . (2) In our paper, we indeed commented on the fact that we unfortunately do not have access to video cameras to identify (and validate) surface processes such as rockfalls and snow avalanches. However, this argument is not valid when it comes to the class of slopequakes which are associated with depth processes and therefore cannot be eye-witnessed. Contrary to Chang et al. (2021), we do not analyse catastrophic landslides, but mainly research minuscule sliding and deformation episodes not visible to the eye. (3) We have evidence that the events we are talking about occur on the slope (more details are given later) - high frequency content and short duration are already some indications. Fischer et al. (2020) also showed and analysed microseismic events generated by a block collapse at Åknes in September 2012. Figure 6b in the manuscript shows the example of a snow avalanche that damaged the cable of one of the geophones.

"Seismic precursors: Schöpa et al. (2018), Dynamics of the Askja caldera July 2014 landslide, Iceland, form seismic signal analysis: precursor, motion and aftermath, Earth Surf. Dynam., 6, 467-485. They presented the precursory seismic signals, source mechanism of landsling, and location of afterslide sequence. A series of numerical modeling was conducted to have a better understanding in mechanism of generation of precursory seismic signals." --> we will add this reference to our manuscript.

"In the current manuscript, actually, there are a lot of statements that are not certainly supported by the results, which are often possible, but they are not open-discussed based on the referred references." --> Please indicate the specific text passages, such that we can review them and take them up in this discussion.

"Seismic interferometry: Kang et al. (2021), Rigidity strengthening of landslide materials measured by seismic interferometry, Remote Sens., 13, 2834. They investigated the temporal link between daily relative velocity changes and in-situ measurements of slope to understand the possible rigidity strengthening." --> this reference will be added. We are aware of the (numerous) works in the field of ambient seismic noise interferometry and listed a few of them in lines 97-101 (before §2.3).

"In this manuscript, a broadband seismometer was deployed in 2009, which is available for analysis of seismic interferometry. Then, tiny velocity variations can be extracted to provide additional information in understanding the physical processes of slopequakes. More discussion should be added in the manuscript." --> (1) The paper deals with the characterization of microseismic signals, which is a different topic. It does not use ambient seismic noise, but everything that is not noise. We do not see where such analysis would fit in the scope of the paper. (2) We acknowledge that seismic interferometry would be interesting for our case study and that is why we mentioned it (lines 97-101). However, there are many factors that currently prevent us from performing such an analysis: (i) unfortunately, the surface geophone network operates in triggered mode, which means we do not have continuous (noise) records. (ii) even if we use the broadband station, it is not possible to do such an analysis with a single station. Only autocorrelation is possible, and the work is currently on-going.

"(2) Line 107 - For the detector of the short-time-average/long-time-average STA/LTA trigger, successful capturing of seismic events depends on proper settings of the trigger parameters: (a) the average values of the absolute amplitude of signal in two consecutive moving-time windows; (b) a pre-set value of the STA/LTA ratio. I cannot find aforementioned parameter settings in the manuscript." --> we used STA = 0.2 s; LTA = 1.9 s, trigger threshold = 4, de-trigger threshold = 1.1. We define an event when the threshold criteria are met on at least 10 channels (i.e. at least 4 different stations since we have 8\*3=24 channels) within a 1 s-long window. These parameters will be added in the manuscript.

" (3) Lines 132-139 - There are two methods for constraining S-wave velocity structure at slope scale: (a) Rayleigh-wave ellipticity from polarization analysis of seismic ambient noise (e.g., H/V spectral ratio)." --> we have already computed H/V spectral ratios for the broadband station. This is on-going work.

"(b) Dispersion curve inferred from the seismic ambient noise cross/auto-correlation functions." --> autocorrelating the records of the broadband station has been part of our planned work for a while.

==> In all cases, such analysis is only possible for the broadband seismometer which is placed in a different and more stable part of the slope (ca. 600 m down from the backscarp which is the fast-moving part). We could not warrant that the potential variations observed there would be representative of the whole slope.

"Thus, a combination of P-wave (seismic reflection profile) and S-wave velocity model is helpful to location source of detected signals. The velocity model is not always difficult to establish." -->

this of course depends on the resolution one wants to achieve. We did not write that we do not have a velocity model. In fact, we have built a 3D velocity model that is derived from both check shots and refraction seismic profiles. However, our overarching goal would be to be able to locate the events accurately enough, such that they delineate the structures at depth, i.e. the cracks and the sliding plane. Since the network has been installed in 2006, many attempts were made to locate the events, but the location uncertainties are too large to be reliable (on the order of 100 m or more for a network where the maximum inter-station distance is 250 m). This can be explained, amongst others, by the small inter-station distances which would require to pick the phases with high precision, but is often complicated by the emergent nature of the signals.

"The second question is related to identification of seismic phases generated by the microseismic events. In fact, it is very difficult to distinguish the P- and S-wave due to the complexity of the source mechanism and the dominance of surface waves in the slopequake-generated seismic signal. However, analysis of three-component seismogram can help us to identify the types of seismic phase, such as the particle motion analysis." --> one of the reasons why the phases are difficult to identify is that the event-station distances are short and the events occur within a relatively shallow layer (max. 100 m or less), hence P- and S-waves, but also surface waves, are likely to arrive almost simultaneously. Polarization analysis does not help in such cases where arrivals overlap and turned out to be rather complex, when we tested the methodology. Besides, the subsurface is very heterogeneous and complex, with a few tens of cm of loose rocks at the surface and fractured to heavily fractured rocks in the first 100 m (which is the area of interest). This entails wave scattering and complex wavefields, and we do not observe clearly linearized particle motion for P-waves for example.

"A possible solution for aforementioned problems is the amplitude source location (ASL) method (Chang et al., 2021), which can estimate the source location by using the large bursts of seismic amplitude without a priori knowledge of velocity model and seismic phase. Such methods, the reader may want to find explored and clarified by the authors." --> In general, while the aforementioned paper is interesting, its goal is rather different since it deals with the location of landslides visible at the surface on a regional scale based on their seismic signatures. In our case, we want to locate microseismic events with an accuracy in the order of a few tens of metres including depth. Nevertheless, we have already explored a similar approach using maximum amplitudes. However, we have many examples in our dataset where the geophone with the maximum amplitude is clearly not the geophone closest to the event (since it features a later arrival time), which we believe is again due to the highly heterogeneous material in which the waves propagate as well as related attenuation and amplification effects. In the paper you mention, it is written that "a well-distributed epicentral distance is a crucial factor for the ASL method. A previous study also demonstrated that the site amplification could strongly influence the location result (Walsh et al., 2017).". In addition, topography is also known to affect amplitudes on large scales (amplification at the mountain tops, de-amplification in the valleys). The difference in elevation between the upper and lower geophone is our network is 100 m; therefore, we expect the topography to add even more complexity.

" (4) Lines 141-146 - At least, the preliminary location of microseismic event is required to support the manually labeling (slopequakes, tremor, rockfall) used in this study. Otherwise, you cannot study the seasonality pattern of microseismic activity, also resulting poor understanding their source mechanisms (origin of slopequake and tremor). Above questions are also existed in the previous study of Provost et al. (2018). You need have some advances compared to Provost et al. (2018)." --> while we completely agree that locating the events would be highly beneficial to our study (and this is why we discuss difficulties in event location on several occasions in the paper), we do not agree that we cannot make sense of the data without. The purpose of our work is precisely to demonstrate that, although event locations are too unstable and inaccurate to further interpret the data, a careful analysis of the waveforms may be a first step towards a better understanding, which is why the classification work was performed. The Provost et al. (2018) paper aimed to propose a global typology of the seismic signals generated by landslides. For that purpose, signals from different sites with different sliding causes and mechanisms were collected, analysed and compared. It turns out that many types of similar signals are observed at different sites: those were named, and their potential physical origin was hypothesized. We used this paper as a basis for our work, although adaptation to our dataset was necessary. Our paper is different from Provost et al. (2018) since it focuses on the entire dataset for a single site. In addition, seasonal acceleration of the displacement in spring at the study site is a well-documented fact (see references). Therefore, it is not particularly surprising to observe that the microseismic activity is also more important in spring. Yet, the classification work that we carried out also emphasizes that some types of signals are more predominant during this increased activity – this is something new that was not known beforehand. Based on these observations and published literature on the topic, we can also start formulating hypotheses on the origins of these events and gaining a better understanding of what our data encompasses. Therefore, the work described in our paper clearly represents a step forward from event detection, even in the absence of event locations.

" (5) Lines 166-169 - In fact, the motion behavior of rockfall is composed of falling, sliding, rolling, and bouncing, which are dominated by the slope angle except for the sliding. For example, as the slope angle is less than 45°, rock mass tends to roll." --> we are not sure how to understand this comment. We drew up a non-exhaustive list of rockfall processes to explain why their seismic signatures can be highly variable. Our purpose in this paper is only to define a general rockfall class., which could eventually be further divided into several sub-classes if of interest.

"(6) Lines 177-179, 294-296 - All records are extracted by using the STA/LTA detector, thus, you cannot have the noise label. As the example of noise signal and spectrograms shown in Fig. A8, there are an obvious and emergent signals with a short duration (the time point from 2 to 4 seconds). I am sure that is not noise signal." --> we agree that there is a signal in Fig. A8. However, there are 3 generators on the slope which do not run continuously, but start only when the batteries of the different instruments are low. Although the STA/LTA generally does not pick up the generator start-up, it happens from time to time as illustrated by Fig. A8. We also wrote (lines 177-178) "In practice, this class only contains few events, since the STA/LTA detector is sufficiently restrictive to discard noisy signals." We think it is still better to have a noise class in which signals of too low quality are sorted and it does not affect the overall results.

"And, that's the reason why the noise label can be easily classified as the source types of regional earthquake (32.3%) and tremor (10.8%) based on the current model (see Figure 5). I am completely not surprised to see above result." --> see line 295: "(...) less than half of the signals corresponding to noise are properly recognised by the CNN. **This is expected, since noise examples are severely under-represented in the training set.**"

"(7) Lines 214-227 - Actually, seismic features observed in this manuscript (such as signal duration, dominant frequency content; see Table 1) can be easily be automatized by machine-learning-based classification/picker/detector approaches." --> yes, we are aware of it, but machine learning is not necessary to extract those features. In the caption of Table 1, we refer to Fig. A18 in the appendix which shows a few probability functions of such features. We did not want to overload the manuscript with too many unnecessary details, but since we tested many different approaches before and in parallel of the CNN implementation, we keep extracting features from the detected events in a more "traditional" way.

"(8) Lines 245-251 - For a specific source (events associated with the slope or surrounding area; Figures A1-A8), there are the clear discrepancies in signal arrival-time, frequency content, and spectral amplitude, which would be useful to constraint the possible source location, further understanding the origin of source mechanisms of microseismic events." --> we do not recognise the move-outs as sufficiently clear on the aforementioned figures since the waveforms are not zoomed-in around the first arrival, although there are of course differences in arrival times. The problem of location is not only dependent on the data, but on the model and assumptions that are made to best explain the observations (e.g. velocity model).

"I suggest that only stacking the three-components seismograms for each station and building the single station-based training model to **detect** the seismic signal individually." --> we are not sure what is meant here. What does "the single station-based training model" refer to? Event detection is successfully performed by STA/LTA.

"In practical real-time application, once the number of triggered station reach a certain threshold (for example, larger than three station), location approach can be implemented." --> see previous comments about location. We define an event when the threshold criteria are met on at least 10 channels (i.e. at least 4 different stations since we have 8\*3=24 channels) within a 1 s-long window. The minimum number of 3 stations required for location is not only valid for real-time applications.

"(9) Line 250 - remove low-quality traces based on a user-defined threshold" Provide a certain definition of a user-defined threshold used in this study." --> The threshold is actually defined automatically as thr = (mean(SNR) - std(SNR)) / max(SNR) where SNR is the signal-to-noise ratio (max/mean) computed for each individual trace. A trace is considered noisy if its SNR is below 0.7 times the threshold (i.e. if it is considered as an

outlier compared to other traces). This will be clarified in the manuscript.

"(10) Figure 4 - The overall flavor of argumentation is somehow "positiveexample"-based, rather than generic and equally balanced. In several cases I can read a lot about the positive outcomes and do not read negative cases. One example is the statement in the section of "Near real-time implementation" and Figure 4." --> Please refer to specific text passages, such that we can review them and take them up in this discussion. Figure 4 is indeed the illustration of an excellent classification and was chosen to illustrate that the classification is unambiguous when in case of clear signals. An example of an erroneous classification is shown in Figure A17 in the appendix. Moreover, the curious and interested reader is kindly invited to click on the link to the webpage where the results are available and judge the classifier by himself. Results are presented in section 4.4 and show an average rate of 80% of good classifications. A large part of the discussion is dedicated to explaining the limits that we see in our work (e.g. only triggered data, no event locations,...). Lastly, the near real-time implementation may not be extraordinary either, yet important to stress because: (1) it provides a quick interpretation of the detected signals to non-seismologists who are in charge of the monitoring of the slope; (2) every newly detected event is assigned to a class which means the interpretation does not rely on a 24/7 seismology service; (3) it provides consistent classes and does not depend on the seismologist's mood-of-the-day and (4) it is a gain of time for the seismologist to QC the automatic classes every morning since the figures are already generated.

"(11) Line 309 - The authors do not have direct evidence to support detected events in this study linked to the rockslope. Current manuscript did not convince me, even though I can feel the enthusiasm of the authors." --> we wonder which kind of evidence is expected by the reviewer, see also our reply to previous comments. There might be a misunderstanding here. We never wrote that we could not locate the events at all. We can of course provide a catalogue with events located on the slope, but with location uncertainties too large to serve as base for any interpretation of event origins – but not as large as doubting that they originate on the slope itself (see Fischer et al. (2020) – events were located at the surface using a homogeneous velocity model). Event location on this slope is notoriously difficult; indeed, several different, independent researchers strove to locate the events using different approaches, but never came to any concluding results. Moreover, we do not see why the reviewer thinks that the events are not related to the slope: all the events we are working with have high dominant frequency contents and are of short duration – these are clear indications that they represent very local events. Besides, due to the heterogeneous and highly attenuative material, for example hammer blows are rarely visible on stations more than 200 m apart from the source. Lastly, although there are now three types of seismic instruments installed on three different parts of the slope, only a handful of the detected events on the surface geophone network are also visible on the borehole and broadband seismometer data. This indicates that the microseismic events are of very local nature.

"(12) Line 317-320 - Please make a systematic comparison between detected catalogue in this study and dataset reported by Fischer et al. (2020). Such comparison, the reader may want to find explored and clarified by the authors." --> Fischer et al. (2020) use the same catalogue as in our study until 2013. The STA/LTA detector parameters have not been changed since 2007.

"(13) Figure 6 - (a) Please add time-series of rainfall, groundwater level, and borehole displacement" --> Rainfall data are plotted in Figure 9e. We unfortunately do not have access to piezometric and borehole displacement data. There is currently a PhD student working with these data together with the microseismic catalogue that is described in our manuscript. We do not think these data are crucial in the scope of this paper.

"(*b*) What's the range of magnitude of regional earthquake? Can the authors predict the peak ground-motion amplitude by using the empirical ground-motion prediction equation? If YES, you can see what's happened in relation between slopequake activity and ground shacking." --> Seismicity in Norway is moderate. Events of magnitude up to 4.5 can occur in the North Sea at about 500 km from the Åknes site. Using the reviewed catalogue of seismicity for the time period from 2010 to now, we found 55 events within a 100 km radius with magnitudes ranging from 1.3 to 3.5. In general, we do not consider earthquake to be a major factor for triggering sliding at Åknes, but it would still be worth investigating more into this direction.

"(14) Lines 325-329 - Seasonality pattern can also be influenced by the detection capacity of network. Snow melting and precipitation can increase the noise level due to the rainfall water dropping to ground and high flow dynamics in river channel and gully at slope (water-generated seismic noise). Please provide more detail discussion about this auestion. Above statement also supports that the number of most of seismic sources show decreasing trend during the summer months (June to September)." --> the question of detectability is addressed in lines 361-363 of the manuscript. Power Spectral Densities (PSD) computed for the broadband seismometer records show some variability of maximum 5 dB at frequencies between 1 and 10 Hz (NB: the geophones placed at the surface are short-period with a 4.5 Hz corner frequency). Lower noise levels are observed in autumn/winter and are higher in spring/summer. Thus, the increased seismicity in spring relatively to the low seismic activity in summer cannot solely be related to variations in the detectability. Besides, as mentioned in one of our previous replies, acceleration of the slope displacement in spring is monitored by other types of instruments and should logically be accompanied by an increase in the seismic activity. Therefore, we are not surprised by our observations.

"(15) Lines 329-337 - Can the authors estimate the changes of forces/pressures caused by the variations in water density? Then, how large magnitude of slopequake can be triggered due to aforementioned state changes?" --> Estimating the stresses resulting from the volumetric expansion of ice in water-filled cracks is a full topic by itself as different processes are involved. Matsuoka & Murton (2008) report theoretical stresses of up to 207 MPa due to ice expansion in cracks at T=-22, but tensile fracturing may occur at stresses as low as 1-2 MPa (Krautblatter et al., 2013). Estimating the largest possible magnitude remains difficult since not all the released energy will be radiated seismically and we would need to have an idea of the fault or crack surface.

"(16) Line 358-361 - This is an unfair estimation. The mean of maximum amplitudes is significantly depended on both of the frequency and distance of wave propagation. Thus, current maximum amplitude used in this study (Figure A14) can be directly used to represent the event size." --> we do not understand this comment. We used the mean of maximum amplitudes as a proxy for event size in the paper. If the comment means that we should have used the maximum amplitude only, then we think it is not necessarily a better estimation since an event that occurs close to one of the geophones could yield a very large amplitude, which is not necessarily representative of the event size. So, both ways have their pros and cons, and we prefer our approach.

"(17) Lines 388-397 - If an additional broadband seismic station far away from the slope sit was included, it is very easy to identify the regional earthquake event or microseismic events on surrounding hillslope." --> yes, we know that, but it has not been done yet. This is partly what we mean when we write that we would like to integrate all data together. It is worth noting that the broadband seismometer was installed about 3 years after the surface geophone network, therefore we cannot rely only on it to classify regional earthquakes. Lastly, while constituting the training set for the "regional" class, we used a reviewed bulletin of regional earthquakes (see line 205) and we checked the broadband seismograms whenever possible.

"(18) Figure A7 - What is an electronic spike? If it is due to the instrumentation problem (data logger), why spike signals can be observed at same time for all stations?" --> actually, spikes are sudden electric transients that occur due to e.g. power outage or storms. This is why they do not have a move-out. We will change the wording from "electronic" to "electric".
* * *
**Summary:**

In our manuscript, we explain the difficulties that we faced in constraining precise event locations, giving concise arguments why we failed, which the reviewer interprets as unproved statement. This is misinterpreted: all "statements" that are made are based on the authors' own experiences and tests. Moreover, it is not a problem to locate the events on the slope – the problem is to locate them with sufficient resolution to highlight structures (faults, cracks, sliding plane) at depth. In this case, the resolution is tightly linked to the knowledge of the velocity model. Although seismic refraction profiles were acquired and inverted to velocities and check shots were made, the study area remains very complex and heterogeneous, perturbing seismic wave propagation at small scales and generating complex wavefields. In such a context, a resolution of a few tens of metres is already coarse and does not allow to accurately locate microseismic events.

The reviewer also points out that velocities and velocity changes could be monitored by taking advantage of the cross-correlation of ambient seismic noise. We are well aware of this method and would surely like to apply it in the future. However, we are currently limited by the data since the surface geophone network (subject of the paper) is operated in triggered (i.e. not continuous) mode. Moreover, although there is indeed a broadband seismometer which records data continuously, it is located in a different, more stable part of the slope, i.e. even if autocorrelation gave a new insight, the results could not be extrapolated to the slope's backscarp which is still the main area of interest.

To conclude, except a few minor changes, we find that most of the comments do not fit the scope of the paper and we believe integrating them would decrease the clarity of the paper. However, we will reformulate some sentences to ensure that descriptions are not misinterpreted. Based on reviewer #2's comments, we made the following changes in the manuscript:

- comment (1): we added references to Kang et al., 2021 and Schöpa et al., 2018;
- comment (2): we added a table containing the STA/LTA computation parameters in the appendix;
- comment (9): we added the specifications on how low-quality traces are discarded.
- comment (18): we replaced all "electronic spike" occurrences by "electric spike".
- general: we reformulated some sentences on our discussion about accurate event location (l. 425 of the revised manuscript).

---

## Author Response (AR2)

**Reply to reviews**

Dear Editor,

We thank you for your careful review of our manuscript.

We took into account all your suggestions which mainly consisted in reformulating some sentences to enhance the manuscript's clarity. Track changes highlight the modifications that we made in the text. In addition, we updated the following figures:

- Figure 1: labels' size in the insets are enlarged;

- Figure 2: event class names are added on top of each subplot;

- Figure 9: smoothed curves computed on a 7-day sliding window are added;

- Figure 10: horizontal bars showing data availability on top of the figure are clarified and new labels are added.

On behalf of all authors,

Nadège Langet